# Fair Canonical Correlation Analysis

**Zhuoping Zhou**[π]**, Davoud Ataee Tarzanagh**[π]**, Bojian Hou**[π]
**Boning Tong, Jia Xu, Yanbo Feng, Qi Long**[σ]**, Li Shen**[σ]
University of Pennsylvania
{zhuopinz@sas.,tarzanaq@,boningt@seas.,jiaxu7@,yanbof@seas.}upenn.edu
{bojian.hou,qlong,li.shen}@pennmedicine.upenn.edu

## Abstract

This paper investigates fairness and bias in Canonical Correlation Analysis (CCA), a widely used statistical technique for examining the relationship between two sets of variables. We present a framework that alleviates unfairness by minimizing the correlation disparity error associated with protected attributes. Our approach enables CCA to learn global projection matrices from all data points while ensuring that these matrices yield comparable correlation levels to group-specific projection matrices. Experimental evaluation on both synthetic and real-world datasets demonstrates the efficacy of our method in reducing correlation disparity error without compromising CCA accuracy.

## 1 Introduction

Canonical Correlation Analysis (CCA) is a multivariate statistical technique that explores the relationship between two sets of variables [30]. Given two datasets $\mathbf{X} \in \mathbb{R}^{N \times D_x}$ and $\mathbf{Y} \in \mathbb{R}^{N \times D_y}$ on the same set of $N$ observations,[1] CCA seeks the $R$–dimensional subspaces where the projections of $\mathbf{X}$ and $\mathbf{Y}$ are maximally correlated, i.e. finds $\mathbf{U} \in \mathbb{R}^{D_x \times R}$ and $\mathbf{V} \in \mathbb{R}^{D_y \times R}$ such that

$$\text{maximize} \ \ \text{trace}\left(\mathbf{U}^\top \mathbf{X}^\top \mathbf{Y} \mathbf{V}\right) \quad \text{subject to} \quad \mathbf{U}^\top \mathbf{X}^\top \mathbf{X} \mathbf{U} = \mathbf{V}^\top \mathbf{Y}^\top \mathbf{Y} \mathbf{V} = \mathbf{I}_R. \qquad \text{(CCA)}$$

CCA finds applications in various fields, including biology [51], neuroscience [2], medicine [79], and engineering [14], for unsupervised or semi-supervised learning. It improves tasks like clustering, classification, and manifold learning by creating meaningful dimensionality-reduced representations [70]. However, CCA can exhibit *unfair* behavior when analyzing data with protected attributes, like sex or race. For instance, in Alzheimer's disease (AD) analysis, CCA can establish correlations between brain imaging and cognitive decline. Yet, if it does not consider the influence of sex, it may result in disparate correlations among different groups because AD affects males and females differently, particularly in cognitive decline [36, 81].

The influence of machine learning on individuals and society has sparked a growing interest in the topic of fairness [42]. While fairness techniques are well-studied in supervised learning [5, 18, 20], attention is shifting to equitable methods in unsupervised learning [11, 12, 15, 34, 49, 55, 64]. Despite extensive work on fairness in machine learning, fair CCA (F-CCA) remains unexplored. This paper investigates F-CCA and introduces new approaches to mitigate bias in (CCA).

For further discussion, we compare CCA with our proposed F-CCA in sample projection, as illustrated in Figure 1. In Figure 1(a), we have samples $\mathbf{x}_1$ and $\mathbf{x}_2$ from matrix $\mathbf{X}$, and in Figure 1(b), their corresponding samples $\mathbf{y}_1$ and $\mathbf{y}_2$ are from matrix $\mathbf{Y}$. CCA learns $\mathbf{U}$ and $\mathbf{V}$ to maximize correlation,

---

[π] Equal contribution  [σ] Corresponding authors
[1] The columns of $\mathbf{X}$ and $\mathbf{Y}$ have been standardized.

37th Conference on Neural Information Processing Systems (NeurIPS 2023).

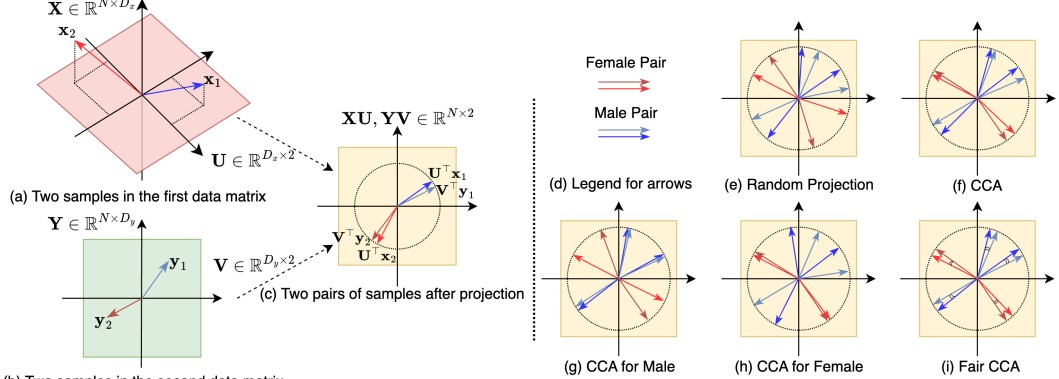

Figure 1: Illustration of CCA and F-CCA, with the sensitive attribute being sex (female and male). Figures (a)–(c) demonstrate the general framework of CCA, while Figures (d)–(i) provide a comparison of the projected results using various strategies. It is important to note that the correlation between two corresponding samples is inversely associated with the angle formed by their projected vectors. F-CCA aims to equalize the average angles among different groups.

inversely related to the angle between the sample vectors. Figure 1(c) demonstrates the proximity within the projected sample pairs $(\mathbf{U}^\top \mathbf{x}_1, \mathbf{V}^\top \mathbf{y}_1)$ and $(\mathbf{U}^\top \mathbf{x}_2, \mathbf{V}^\top \mathbf{y}_2)$. In Figure 1(d)-(i), we compare the results of different learning strategies. There are five pairs of samples, with female pairs highlighted in red and male pairs shown in blue. Random projection (Figure 1(e)) leads to randomly large angles between corresponding sample vectors. CCA reduces angles compared to random projection (Figure 1(f)), but significant angle differences between male and female pairs indicate bias. Using sex-based projection matrices heavily biases the final projection, favoring one sex over the other (Figures 1(g) and 1(h)). To address this bias, our F-CCA maximizes correlation within pairs and ensures equal correlations across different groups, such as males and females (Figure 1(i)). Note that while this illustration represents individual fairness, the desired outcome in practice is achieving similar average angles for different groups.

**Contributions.** This paper makes the following key contributions:

- We introduce fair CCA (F-CCA), a model that addresses fairness issues in (CCA) by considering multiple groups and minimizing the correlation disparity error of protected attributes. F-CCA aims to learn global projection matrices from all data points while ensuring that these projection matrices produce a similar amount of correlation as group-specific projection matrices.

- We propose two optimization frameworks for F-CCA: multi-objective and single-objective. The multi-objective framework provides an automatic trade-off between global correlation and equality in group-specific correlation disparity errors. The single-objective framework offers a simple approach to approximate fairness in CCA while maintaining a strong global correlation, requiring a tuning parameter to balance these objectives.

- We develop a gradient descent algorithm on the generalized Stiefel manifold to solve the multi-objective problem, with convergence guarantees to a Pareto stationary point. This approach extends Riemannian gradient descent [8, 9] to multi-objective optimization, accommodating a broader range of retraction maps than exponential retraction [23, 6]. Furthermore, we provide a similar algorithm for single-objective problems, also with convergence guarantees to a stationary point.

- We provide extensive empirical results showcasing the efficacy of the proposed algorithms. Comparison against the CCA method on synthetic and real datasets highlights the benefits of the F-CCA approach, validating the theoretical findings [2].

**Organization:** Section 2 covers related work. Our proposed approach is detailed in Section 3, along with its theoretical guarantees. Section 4 showcases numerical experiments, while Section 5 discusses implications and future research directions.

---

[2]Code is available at https://github.com/PennShenLab/Fair_CCA.

## 2   Related work

**Canonical Correlation Analysis (CCA).** CCA was first introduced by [28, 29]. Since then, it has been utilized to explore relations between variables in various fields of science, including economics [72], psychology [19, 27], geography [45], medicine [39], physics [76], chemistry [69], biology [62], time-series modeling [26], and signal processing [57]. Recently, CCA has demonstrated its applicability in modern fields of science such as neuroscience, machine learning, and bioinformatics [59, 60]. CCA has been used to explore relations for developing brain-computer interfaces [10, 46] and in the field of imaging genetics [22]. CCA has also been applied for feature selection [47], feature extraction and fusion [61], and dimension reduction [71]. Additionally, numerous studies have applied CCA in bioinformatics and computational biology, such as [54, 56, 58]. The broad range of application domains highlights the versatility of CCA in extracting relations between variables, making it a valuable tool in scientific research.

**Fairness.** Fairness in machine learning has been a growing area of research, with much of the work focusing on fair supervised methods [5, 16, 18, 20, 67, 78]. However, there has also been increasing attention on fair methods for unsupervised learning tasks [11, 12, 15, 34, 33, 49, 55, 64, 50, 66]. In particular, Samadi et al. [55] proposed a semi-definite programming approach to ensure fairness in PCA. Kleindessner et al. [33, 34] focused on fair PCA formulation for multiple groups and proposed a kernel-based fair PCA. Kamani et al. [32] introduced an efficient gradient method for fair PCA, addressing multi-objective optimization. In this paper, we propose a novel multi-objective framework for F-CCA, converting constrained F-CCA problems to unconstrained ones on a generalized Riemannian manifold. This framework enables the adaptation of efficient gradient techniques for numerical optimization on Riemannian manifolds.

**Riemannian Optimization.** Riemannian optimization extends Euclidean optimization to smooth manifolds, enabling the minimization of $f(\mathbf{x})$ on a Riemannian manifold $\mathcal{M}$ and converting constrained problems into unconstrained ones [1, 8]. It finds applications in various domains such as matrix/tensor factorization [31, 63], PCA [21], and CCA [77]. Specifically, CCA can be formulated as Riemannian optimization on the Stiefel manifold [13, 43]. In our work, we utilize Riemannian optimization to develop a multi-objective framework for F-CCAs on generalized Stiefel manifolds.

## 3   Fair Canonical Correlation Analysis

This section introduces the formulation and optimization algorithms for F-CCA.

### 3.1   Preliminary

Real numbers are represented as $\mathbb{R}$, with $\mathbb{R}_+$ for nonnegative values and $\mathbb{R}_{++}$ for positives. Vectors and matrices use bold lowercase and uppercase letters (e.g., $\mathbf{a}$, $\mathbf{A}$) with elements $a_i$ and $a_{ij}$. For $\mathbf{x}, \mathbf{y} \in \mathbb{R}^m$, $\mathbf{x} \prec \mathbf{y}$ and $\mathbf{x} \preceq \mathbf{y}$ mean $\mathbf{y} - \mathbf{x} \in \mathbb{R}^m_{++}$ and $\mathbf{y} - \mathbf{x} \in \mathbb{R}^m_+$, respectively. For a symmetric matrix $\mathbf{A} \in \mathbb{R}^{N \times N}$, $\mathbf{A} \succ 0$ and $\mathbf{A} \succeq 0$ denote positive definiteness and positive semidefiniteness (PSD), respectively. $\mathbf{I}_D$, $\mathbf{J}_D$, and $\mathbf{0}_D$ are $D \times D$ identity, all-ones, and all-zeros matrices. $\Lambda_i(\mathbf{A})$ stands for the $i$-th singular values of $\mathbf{A}$. Matrix norms are defined as $\|\mathbf{A}\|_1 = \sum_{ij} |a_{ij}|$, $\|\mathbf{A}\| = \max_i \Lambda_i(\mathbf{A})$, and $\|\mathbf{A}\|_F := (\sum_{ij} |a_{ij}|^2)^{1/2}$. We introduce some preliminaries on manifold optimization [1, 6, 8]. Given a PSD matrix $\mathbf{B} \in \mathbb{R}^{D \times D}$, the generalized Stiefel manifold is defined as

$$\mathtt{St}(D, R, \mathbf{B}) = \left\{ \mathbf{Z} \in \mathbb{R}^{D \times R} \mid \mathbf{Z}^\top \mathbf{B} \mathbf{Z} = \mathbf{I}_R \right\}. \tag{1}$$

The tangent space of the manifold $\mathcal{M} = \mathtt{St}(D, R, \mathbf{B})$ at $\mathbf{Z} \in \mathcal{M}$ is given by

$$\mathcal{T}_\mathbf{Z}\mathcal{M} = \left\{ \mathbf{W} \in \mathbb{R}^{D \times R} \mid \mathbf{Z}^\top \mathbf{B} \mathbf{W} + \mathbf{W}^\top \mathbf{B} \mathbf{Z} = \mathbf{0}_R \right\}. \tag{2}$$

The tangent bundle of a smooth manifold $\mathcal{M}$, which consists of $\mathcal{T}_\mathbf{Z}\mathcal{M}$ at all $\mathbf{Z} \in \mathcal{M}$, is defined as

$$\mathcal{T}\mathcal{M} = \left\{ (\mathbf{Z}, \mathbf{W}) \mid \mathbf{Z} \in \mathcal{M}, \ \mathbf{W} \in \mathcal{T}_\mathbf{Z}\mathcal{M} \right\}. \tag{3}$$

**Definition 1.** *A retraction on a differentiable manifold $\mathcal{M}$ is a smooth mapping from its tangent bundle $\mathcal{T}\mathcal{M}$ to $\mathcal{M}$ that satisfies the following conditions, with $R^\mathbf{z}$ being the retraction of $R$ to $\mathcal{T}_\mathbf{Z}\mathcal{M}$:*

*1. $R^\mathbf{z}(\mathbf{0}) = \mathbf{Z}$, for all $\mathbf{Z} \in \mathcal{M}$, where $\mathbf{0}$ denotes the zero element of $\mathcal{T}_\mathbf{Z}\mathcal{M}$.*

2. *For any* $\mathbf{Z} \in \mathcal{M}$, *it holds that* $\lim_{\mathcal{T}_\mathbf{Z} \mathcal{M} \ni \boldsymbol{\xi} \to 0} \frac{\|R^\mathbf{z}(\boldsymbol{\xi}) - (\mathbf{Z} + \boldsymbol{\xi})\|_F}{\|\boldsymbol{\xi}\|_F} = 0$.

In the numerical experiments, this work employs a generalized polar decomposition-based retraction. Given a PSD matrix $\mathbf{B} \in \mathbb{R}^{D \times D}$, for any $\boldsymbol{\xi} \in \mathcal{T}_\mathbf{Z} \mathcal{M}$ with $\mathcal{M} = \mathrm{St}(D, R, \mathbf{B})$, it is defined as:

$$R^\mathbf{z}(\boldsymbol{\xi}) = \bar{\mathbf{U}}(\mathbf{Q}\mathbf{\Lambda}^{-\frac{1}{2}}\mathbf{Q}^\top)\bar{\mathbf{V}}^\top, \tag{4}$$

where $\bar{\mathbf{U}}\Sigma\bar{\mathbf{V}}^\top = \boldsymbol{\xi}$ is the singular value decomposition of $\boldsymbol{\xi}$, and $\mathbf{Q}, \mathbf{\Lambda}$ are obtained from the eigenvalue decomposition $\mathbf{Q}\mathbf{\Lambda}Q^\top = \bar{\mathbf{U}}^\top\mathbf{B}\bar{\mathbf{U}}$. Further details on retraction choices are in Appendix A.1.

### 3.2 Correlation Disparity Error

As previously mentioned, applying CCA to the entire dataset could lead to a biased result, as some groups might dominate the analysis while others are overlooked. To avoid this, we can perform CCA separately on each group and compare the results. Indeed, we can compare the performance of CCA on each group's data with the performance of CCA on the whole dataset, which includes all groups' data. The goal is to find a balance between the benefits and sacrifices of different groups so that each group's contribution to the CCA analysis is treated fairly. In particular, suppose the datasets $\mathbf{X} \in \mathbb{R}^{N \times D_x}$ and $\mathbf{Y} \in \mathbb{R}^{N \times D_y}$ on the same set of $N$ observations, belong to $K$ different groups $\{(\mathbf{X}^k, \mathbf{Y}^k)\}_{k=1}^K$ with $\mathbf{X}^k \in \mathbb{R}^{N_k \times D_x}$ and $\mathbf{Y}^k \in \mathbb{R}^{N_k \times D_y}$, based on demographics or some other semantically meaningful clustering. These groups need not be mutually exclusive; each group can be defined as a different weighting of the data.

To determine how each group is affected by F-CCA, we can compare the structure learned from each group's data $(\mathbf{X}^k, \mathbf{Y}^k)$ with the structure learned from all groups' data combined $(\mathbf{X}, \mathbf{Y})$. A fair CCA approach seeks to balance the benefits and drawbacks of each group's contribution to the analysis. Specifically, if we train global subspaces $\mathbf{U} \in \mathbb{R}^{D_x \times R}$ and $\mathbf{V} \in \mathbb{R}^{D_y \times R}$ on $k$-th group dataset $(\mathbf{X}^k, \mathbf{Y}^k)$, we can identify the group-specific (local) weights represented by $(\mathbf{U}^k, \mathbf{V}^k)$ that has the best performance on that dataset. Thus, F-CCA algorithm should be able to learn global weights $(\mathbf{U}, \mathbf{V})$ on all data points while ensuring that each group's correlation on the CCA learned by the whole dataset is equivalent to the group-specific subspaces learned only by its own data.

To define these fairness criteria, we introduce correlation disparity error as follows:

**Definition 2** (**Correlation Disparity Error**). *Consider a pair of datasets* $(\mathbf{X}, \mathbf{Y})$ *with* $K$ *sensitive groups with data matrix* $\{(\mathbf{X}^k, \mathbf{Y}^k)\}_{k=1}^K$ *representing each sensitive group's data samples. Then, for any* $(\mathbf{U}, \mathbf{V})$, *the correlation disparity error for each sensitive group* $k \in [K]$ *is defined as*

$$\mathcal{E}^k(\mathbf{U}, \mathbf{V}) := \mathrm{trace}\left(\mathbf{U}^{k,\star^\top}\mathbf{X}^{k^\top}\mathbf{Y}^k\mathbf{V}^{k,\star}\right) - \mathrm{trace}\left(\mathbf{U}^\top\mathbf{X}^{k^\top}\mathbf{Y}^k\mathbf{V}\right), \qquad 1 \le k \le K. \tag{5}$$

*Here,* $(\mathbf{U}^{k,\star}, \mathbf{V}^{k,\star})$ *is the maximizer of the following group-specific CCA problem:*

$$\text{maximize} \quad \mathrm{trace}\left(\mathbf{U}^{k^\top}\mathbf{X}^{k^\top}\mathbf{Y}^k\mathbf{V}^k\right) \quad \text{subj. to} \quad \mathbf{U}^{k^\top}\mathbf{X}^{k^\top}\mathbf{X}^k\mathbf{U}^k = \mathbf{V}^{k^\top}\mathbf{Y}^{k^\top}\mathbf{Y}^k\mathbf{V}^k = \mathbf{I}_R. \tag{6}$$

This measure shows how much correlation we are suffering for any global $(\mathbf{U}, \mathbf{V})$, with respect to the loss of optimal local $(\mathbf{U}^{k,\star}, \mathbf{V}^{k,\star})$ that we can learn based on data points $(\mathbf{X}^k, \mathbf{Y}^k)$.

Using Definition 2, we can define F-CCA as follows:

**Definition 3** (**Fair CCA**). *A CCA pair* $(\mathbf{U}^\star, \mathbf{V}^\star)$ *is called fair if the correlation disparity error among* $K$ *different groups is equal, i.e.,*

$$\mathcal{E}^k(\mathbf{U}^\star, \mathbf{V}^\star) = \mathcal{E}^s(\mathbf{U}^\star, \mathbf{V}^\star), \qquad \forall k \ne s, \quad k, s \in [K]. \tag{7}$$

*A CCA pair* $(\mathbf{U}^\star, \mathbf{V}^\star)$ *that achieves the same disparity error for all groups is called a fair CCA.*

Next, we introduce the concept of pairwise correlation disparity error for CCA, which measures the variation in correlation disparity among different groups.

**Definition 4** (**Pairwise Correlation Disparity Error**). *The pairwise correlation disparity error for any global* $(\mathbf{U}, \mathbf{V})$ *and group-specific subspaces* $\{(\mathbf{U}^{k,\star}, \mathbf{V}^{k,\star})\}_{k=1}^K$, *is defined as*

$$\Delta^{k,s}(\mathbf{U}, \mathbf{V}) := \phi\left(\mathcal{E}^k(\mathbf{U}, \mathbf{V}) - \mathcal{E}^s(\mathbf{U}, \mathbf{V})\right), \qquad \forall k \ne s, \quad k, s \in [K]. \tag{8}$$

*Here,* $\phi : \mathbb{R} \to \mathbb{R}_+$ *is a penalty function such as* $\phi(x) = \exp(x)$, $\phi(x) = x^2$, *or* $\phi(x) = |x|$.

| **Algorithm 1**: A Multi-Objective Gradient Method for F-CCA (MF-CCA) | **Algorithm 2**: A Single-Objective Gradient Method for F-CCA (SF-CCA) |
|---|---|
| 1: **Input**: $(\mathbf{X}, \mathbf{Y})$, $(\mathbf{U}_0, \mathbf{V}_0)$, $(R, T) \in \mathbb{N} \times \mathbb{N}$; and stepsizes $\{(\eta_t^{\mathbf{u}}, \eta_t^{\mathbf{v}})\}_{t=1}^T$; | 1: **Input**: $(\mathbf{X}, \mathbf{Y})$, $(\mathbf{U}_0, \mathbf{V}_0)$, $(R, T) \in \mathbb{N} \times \mathbb{N}$; $\lambda \in \mathbb{R}_{++}$; and stepsizes $\{(\eta_t^{\mathbf{u}}, \eta_t^{\mathbf{v}})\}_{t=1}^T$; |
| 2: Find the $R$-rank subspaces for each group, $\{(\mathbf{U}^{k,*}, \mathbf{V}^{k,*})\}_{k=1}^K$ using (6). | 2: Find the $R$-rank subspaces for each group, $\{(\mathbf{U}^{k,*}, \mathbf{V}^{k,*})\}_{k=1}^K$ using (6). |
| 3: **for** $t = 0, \ldots, T-1$ **do** | 3: **for** $t = 0, \ldots, T-1$ **do** |
| 4:    Find $(\mathbf{P}_t^{\mathbf{u}}, \mathbf{P}_t^{\mathbf{v}})$ by solving (10). | 4:    Find $(\mathbf{G}_t^{\mathbf{u}}, \mathbf{G}_t^{\mathbf{v}})$ by solving (12). |
| 5:    $\mathbf{U}_{t+1} \leftarrow R^{\mathbf{u}}(\eta_t^{\mathbf{u}} \mathbf{P}_t^{\mathbf{u}})$ | 5:    $\mathbf{U}_{t+1} \leftarrow R^{\mathbf{u}}(\eta_t^{\mathbf{u}} \mathbf{G}_t^{\mathbf{u}})$ |
| 6:    $\mathbf{V}_{t+1} \leftarrow R^{\mathbf{v}}(\eta_t^{\mathbf{v}} \mathbf{P}_t^{\mathbf{v}})$ | 6:    $\mathbf{V}_{t+1} \leftarrow R^{\mathbf{v}}(\eta_t^{\mathbf{v}} \mathbf{G}_t^{\mathbf{v}})$ |
| 7: **end for** | 7: **end for** |
| 8: **Output**: $(\mathbf{U}_T, \mathbf{V}_T)$, $\{(\mathbf{U}^{k,*}, \mathbf{V}^{k,*})\}_{k=1}^K$. | 8: **Output**: $(\mathbf{U}_T, \mathbf{V}_T)$, $\{(\mathbf{U}^{k,*}, \mathbf{V}^{k,*})\}_{k=1}^K$. |

The motivation for incorporating pairwise correlation disparity error in our approach can be attributed to the work by [40, 55] in the context of PCA. To facilitate convergence analysis, we will primarily consider smooth penalization functions, such as squared or exponential penalties.

### 3.3   A Multi-Objective Framework for Fair CCA

In this section, we introduce an optimization framework for balancing correlation and disparity errors. Let $f_1(\mathbf{U}, \mathbf{V}) := -\text{trace}\left(\mathbf{U}^\top \mathbf{X}^\top \mathbf{Y} \mathbf{V}\right)$, $f_2(\mathbf{U}, \mathbf{V}) := \Delta^{1,2}(\mathbf{U}, \mathbf{V}), \ldots, f_M(\mathbf{U}, \mathbf{V}) := \Delta^{K-1,K}(\mathbf{U}, \mathbf{V})$. The optimization problem of finding an optimal Pareto point of $\mathbf{F}$ is denoted by

$$
\begin{aligned}
\underset{\mathbf{U}, \mathbf{V}}{\text{minimize}} \quad & \mathbf{F}(\mathbf{U}, \mathbf{V}) := [f_1(\mathbf{U}, \mathbf{V}), f_2(\mathbf{U}, \mathbf{V}), \ldots, f_M(\mathbf{U}, \mathbf{V})], \\
\text{subj. to} \quad & \mathbf{U} \in \mathcal{U}, \quad \mathbf{V} \in \mathcal{V},
\end{aligned}
\tag{9}
$$

where $\mathcal{U} := \{\mathbf{U} \in \mathbb{R}^{D_x \times R} | \mathbf{U}^\top \mathbf{X}^\top \mathbf{X} \mathbf{U} = \mathbf{I}_R\}$ and $\mathcal{V} := \{\mathbf{V} \in \mathbb{R}^{D_y \times R} | \mathbf{V}^\top \mathbf{Y}^\top \mathbf{Y} \mathbf{V} = \mathbf{I}_R\}$.

A point $(\mathbf{U}, \mathbf{V}) \in \mathcal{U} \times \mathcal{V}$ satisfying $\text{Im}(\nabla \mathbf{F}(\mathbf{U}, \mathbf{V})) \cap (-\mathbb{R}_{++}^M) = \emptyset$ is called *critical Pareto*. Here, $\text{Im}$ denotes the image of Jacobian of $\mathbf{F}$. An *optimum Pareto point* of $\mathbf{F}$ is a point $(\mathbf{U}^\star, \mathbf{V}^\star) \in \mathcal{U} \times \mathcal{V}$ such that there exists no other $(\mathbf{U}, \mathbf{V}) \in \mathcal{U} \times \mathcal{V}$ with $\mathbf{F}(\mathbf{U}, \mathbf{V}) \prec \mathbf{F}(\mathbf{U}^\star, \mathbf{V}^\star)$. Moreover, a point $\mathbf{U}^\star, \mathbf{V}^\star \in \mathcal{U} \times \mathcal{V}$ is a *weak optimal Pareto* of $\mathbf{F}$ if there is no $(\mathbf{U}, \mathbf{V}) \in \mathcal{U} \times \mathcal{V}$ with $\mathbf{F}(\mathbf{U}, \mathbf{V}) \preceq \mathbf{F}(\mathbf{U}^\star, \mathbf{V}^\star)$. The multi-objective framework (9) addresses the challenge of handling conflicting objectives and achieving optimal trade-offs between them.

To effectively solve Problem (9), we propose utilizing a gradient descent method on the manifold $\mathcal{U} \times \mathcal{V}$ that ensures convergence to a *Pareto stationary point*. The proposed gradient descent algorithm for solving (9) is provided in **Algorithm 1**. For each $(\mathbf{U}, \mathbf{V}) \in \mathcal{U} \times \mathcal{V}$, let $\mathbf{P} := (\mathbf{P}^{\mathbf{u}}, \mathbf{P}^{\mathbf{v}})$ with $\mathbf{P}^{\mathbf{u}} \in \mathcal{T}_{\mathbf{U}} \mathcal{U}$ and $\mathbf{P}^{\mathbf{v}} \in \mathcal{T}_{\mathbf{V}} \mathcal{V}$. The iterates $(\mathbf{P}_t^{\mathbf{u}}, \mathbf{P}_t^{\mathbf{v}})$ in Step 4 are obtained by solving the following subproblem in the joint tangent plane $\mathcal{T}_{\mathbf{U}} \mathcal{U} \times \mathcal{T}_{\mathbf{V}} \mathcal{V}$:

$$
\min_{\mathbf{P} \in \mathcal{T}_{\mathbf{U}} \mathcal{U} \times \mathcal{T}_{\mathbf{V}} \mathcal{V}} Q_t(\mathbf{P}), \quad \text{where } Q_t(\mathbf{P}) := \left\{ \max_{i \in [M]} \text{trace}\left(\mathbf{P}^\top \nabla f_i((\mathbf{U}_t, \mathbf{V}_t))\right) + \frac{1}{2} \|\mathbf{P}\|_{\text{F}}^2 \right\}. \tag{10}
$$

If $(\mathbf{U}_t, \mathbf{V}_t) \notin \mathcal{U} \times \mathcal{V}$ is not a Pareto stationary point, Problem (10) has a unique nonzero solution $\mathbf{P}_t$ (see Lemma 7), known as the *steepest descent direction* for $\mathbf{F}$ at $(\mathbf{U}_t, \mathbf{V}_t)$. In Steps 5 and 6, $R^{\mathbf{u}}$ and $R^{\mathbf{v}}$ denote the retractions onto the tangent spaces $\mathcal{T}_{\mathbf{U}} \mathcal{U}$ and $\mathcal{T}_{\mathbf{V}} \mathcal{V}$, respectively; refer to Definition 1.

**Assumption A.** *For a given subset $\mathcal{S}$ of the tangent bundle $\mathcal{T}\mathcal{U} \times \mathcal{T}\mathcal{V}$, there exists a constant $L_F$ such that, for all $(\mathbf{Z}, \mathbf{P}) \in \mathcal{S}$, we have $\mathbf{F}(R^{\mathbf{z}}(\mathbf{P})) \preceq \mathbf{F}(\mathbf{Z}) + \nabla + (L_F/2)\|\mathbf{P}\|_{\text{F}}^2 \mathbf{1}_M$, where $\nabla_i := \langle \nabla f_i(\mathbf{Z}), \mathbf{P}\rangle$, $\nabla := [\nabla_1, \cdots, \nabla_M]^\top \in \mathbb{R}^M$, and $R^{\mathbf{z}}$ is the retraction.*

The above assumption extends [8, A 4.3] to multi-objective optimization, and it always holds for the *exponential* map (exponential retraction) if the gradient of $\mathbf{F}$ is $L_F$-Lipschitz continuous [23, 6].

**Theorem 5.** *Suppose Assumption A holds. Let $(\mathbf{U}_t, \mathbf{V}_t)$ be the sequence generated by MF-CCA. Let $f_i^* := \inf\{f_i(\mathbf{U}, \mathbf{V}) : (\mathbf{U}, \mathbf{V}) \in \mathcal{U} \times \mathcal{V}\}$, for all $i \in [M]$ and define $f_{i_*}(\mathbf{U}_0, \mathbf{V}_0) - f_{i_*}^* := \min\{f_i(\mathbf{U}_0, \mathbf{V}_0) - f_i^* : i \in [M]\}$. If $\eta_t^{\mathbf{u}} = \eta_t^{\mathbf{v}} = \eta \leq 1/L_F$ for all $t \in \{0, \ldots, T-1\}$, then*

$$
\min\left\{ \|\mathbf{P}_t\|_{\text{F}} : t = 0, \ldots, T-1 \right\} \leq \frac{2}{\eta} \left[ \frac{f_{i_*}(\mathbf{U}_0, \mathbf{V}_0) - f_{i_*}^*}{T} \right]^{\frac{1}{2}}.
$$

*Proof Sketch.* We employ Lemma 7 to establish the unique solution $\mathbf{P}_t$ for subproblem (10). Lemmas 9 and 10 provide estimates for the decrease of function $\mathbf{F}$ along $\mathbf{P}_t$: For any $\eta_t \geq 0$, we have $\mathbf{F}(\mathbf{U}_{t+1}, \mathbf{V}_{t+1}) \preceq \mathbf{F}(\mathbf{U}_t, \mathbf{V}_t) - (\eta_t - L_F \eta_t^2/2) \|\mathbf{P}_t\|_{\mathrm{F}}^2 \mathbf{1}_M$. Summing this inequality over $t = 0, 1, \ldots, T-1$ and applying our step size condition yields the desired result. $\qquad\square$

Theorem 5 provides a generalization of [8, Corollary 4.9] to the multi-objective optimization, showing that the norm of Pareto descent directions converges to zero. Consequently, the solutions produced by the algorithm converge to a stationary fair subspace. It is worth mentioning that multi-objective optimization in [23, 6] relies on the Riemannian exponential map, whereas the above theorem covers broader (and practical) retraction maps.

### 3.4 A Single-Objective Framework for Fair CCA

In this section, we introduce a straightforward and effective single-objective framework. This approach simplifies F-CCA optimization, lowers computational requirements, and allows for fine-tuning fairness-accuracy trade-offs using the hyperparameter $\lambda$. Specifically, by employing a regularization parameter $\lambda > 0$, our proposed fairness model for F-CCA is expressed as follows:

$$
\begin{aligned}
\underset{\mathbf{U}, \mathbf{V}}{\text{minimize}} \quad & f(\mathbf{U}, \mathbf{V}) := -\operatorname{trace}\left(\mathbf{U}^\top \mathbf{X}^\top \mathbf{Y} \mathbf{V}\right) + \lambda \Delta\left(\mathbf{U}, \mathbf{V}\right), \\
\text{subj. to} \quad & \mathbf{U} \in \mathcal{U}, \quad \mathbf{V} \in \mathcal{V},
\end{aligned}
\tag{11}
$$

where $\Delta\left(\mathbf{U}, \mathbf{V}\right) = \sum_{i,j \in [K], i \neq j} \Delta^{i,j}\left(\mathbf{U}, \mathbf{V}\right)$; see Definiton 4.

The choice of $\lambda$ in the model determines the emphasis placed on different objectives. When $\lambda$ is large, the model prioritizes fairness over minimizing subgroup errors. Conversely, if $\lambda$ is small, the focus shifts towards minimizing subgroup correlation errors rather than achieving perfect fairness. In other words, it is possible to obtain perfectly F-CCA subspaces; however, this may come at the expense of larger errors within the subgroups. The constant $\lambda$ in the model allows for a flexible trade-off between fairness and minimizing subgroup correlation errors, enabling us to find a balance based on the specific requirements and priorities of the problem at hand.

The proposed gradient descent algorithm for solving (11) is provided as **Algorithm 2**. For each $(\mathbf{U}, \mathbf{V}) \in \mathcal{U} \times \mathcal{V}$, let $\mathbf{G} := (\mathbf{G}^{\mathbf{u}}, \mathbf{G}^{\mathbf{v}})$ with $\mathbf{G}^{\mathbf{u}} \in \mathcal{T}_{\mathbf{U}} \mathcal{U}$ and $\mathbf{G}^{\mathbf{v}} \in \mathcal{T}_{\mathbf{V}} \mathcal{V}$. The iterates $(\mathbf{G}_t^{\mathbf{u}}, \mathbf{G}_t^{\mathbf{v}})$ are obtained by solving the following problem in the joint tangent plane $\mathcal{T}_{\mathbf{U}} \mathcal{U} \times \mathcal{T}_{\mathbf{V}} \mathcal{V}$:

$$
\min_{\mathbf{G} \in \mathcal{T}_{\mathbf{U}} \mathcal{U} \times \mathcal{T}_{\mathbf{V}} \mathcal{V}} q_t(\mathbf{G}), \quad \text{where} \quad q_t(\mathbf{G}) := \left\{ \operatorname{trace}\left(\mathbf{G}^\top \nabla f((\mathbf{U}_t, \mathbf{V}_t))\right) + \frac{1}{2}\|\mathbf{G}\|_{\mathrm{F}}^2 \right\}. \tag{12}
$$

The solutions $(\mathbf{G}_t^{\mathbf{u}}, \mathbf{G}_t^{\mathbf{v}})$ are maintained on the manifolds using the retraction operations $R^{\mathbf{u}}$ and $R^{\mathbf{v}}$.

**Assumption B.** *For a subset $\mathcal{S} \subseteq \mathcal{T}\mathcal{U} \times \mathcal{T}\mathcal{V}$, there exists a constant $L_f$ such that for all $(\mathbf{Z}, \mathbf{G}) \in \mathcal{S}$, $f(R^{\mathbf{z}}(\mathbf{G})) \leq \mathbf{F}(\mathbf{Z}) + \langle \nabla f(\mathbf{Z}), \mathbf{G} \rangle + (L_f/2) \|\mathbf{G}\|_{\mathrm{F}}^2$, with $R^{\mathbf{z}}$ as the retraction.*

**Theorem 6.** *Suppose Assumption B holds. Let $(\mathbf{U}_t, \mathbf{V}_t)$ be the sequence generated by SF-CCA. Let $f^* := \inf\{f(\mathbf{U}, \mathbf{V}) : (\mathbf{U}, \mathbf{V}) \in \mathcal{U} \times \mathcal{V}\}$. If $\eta_t^{\mathbf{u}} = \eta_t^{\mathbf{v}} = \eta \leq 1/L_f$ for all $t \in [T]$, then*

$$
\min\left\{ \|\mathbf{G}_t\|_{\mathrm{F}} : t = 0, \ldots, T-1 \right\} \leq \frac{2}{\eta} \left[ \frac{f(\mathbf{U}_0, \mathbf{V}_0) - f^*}{T} \right]^{\frac{1}{2}}.
$$

**Comparison between MF-CCA and SF-CCA:** MF-CCA addresses conflicting objectives and achieves optimal trade-offs automatically, but it necessitates the inclusion of $\binom{K}{2}$ additional objectives. SF-CCA, on the other hand, provides a simpler approach but requires tuning an extra hyperparameter $\lambda$. When choosing between the two methods, it is crucial to consider the trade-off between complexity and simplicity, as well as the number of objectives and the need for hyperparameter tuning.

## 4 Experiments

In this section, we provide empirical results showcasing the efficacy of the proposed algorithms.

## 4.1 Evaluation Criteria and Selection of Tuning Parameter

F-CCA's performance is evaluated on correlation and fairness for each dimension of subspaces. Let $\mathbf{U} = [\mathbf{u}_1, \cdots, \mathbf{u}_R] \in \mathbb{R}^{D_x \times R}$ and $\mathbf{V} = [\mathbf{v}_1, \cdots, \mathbf{v}_R] \in \mathbb{R}^{D_y \times R}$. The $r$-th canonical correlation is defined as follows:

$$\rho_r = \frac{\mathbf{u}_r^\top \mathbf{X}^\top \mathbf{Y} \mathbf{v}_r}{\sqrt{\mathbf{u}_r^\top \mathbf{X}^\top \mathbf{X} \mathbf{u}_r \mathbf{v}_r^\top \mathbf{Y}^\top \mathbf{Y} \mathbf{v}_r}}, \quad r = 1, \ldots, R. \tag{13a}$$

Next, in terms of fairness, we establish the following two key measures:

$$\Delta_{\max,r} = \max_{i,j \in [K]} |\mathcal{E}^i(\mathbf{u}_r, \mathbf{v}_r) - \mathcal{E}^j(\mathbf{u}_r, \mathbf{v}_r)|, \quad r = 1, \ldots, R, \tag{13b}$$

$$\Delta_{\text{sum},r} = \sum_{i,j \in [K]} |\mathcal{E}^i(\mathbf{u}_r, \mathbf{v}_r) - \mathcal{E}^j(\mathbf{u}_r, \mathbf{v}_r)|, \quad r = 1, \ldots, R. \tag{13c}$$

Here, $\Delta_{\max,r}$ measures maximum disparity error, while $\Delta_{\text{sum},r}$ represents aggregate disparity error. The aim is to reach $\Delta_{\max,r}$ and $\Delta_{\text{sum},r}$ of 0 without sacrificing correlation ($\rho_r$) compared to CCA. We conduct a detailed analysis using component-wise measurements (13) instead of matrix versions; for more discussions, see Appendix C.2.

The `canoncorr` function from MATLAB and [35] is used to solve (CCA). For MF-CCA and SF-CCA, the learning rate is searched on a grid in $\{1e-1, 5e-2, 1e-2, \ldots, 1e-5\}$, and for SF-CCA, $\lambda$ is searched on a grid in $\{1e-2, 1e-1, 0.5, 1, 2, \ldots, 10\}$. Sensitivity analysis of $\lambda$ is provided in Appendix B.2. The learning rate decreases with the square root of the iteration number. Termination of algorithms occurs when the descent direction norm is below $1e-4$.

## 4.2 Dataset

### 4.2.1 Synthetic Data

Following [4, 44], our synthetic data are generated using the Gaussian distribution

$$\begin{pmatrix} \mathbf{X} \\ \mathbf{Y} \end{pmatrix} \sim N \left( \begin{bmatrix} \mu_\mathbf{X} \\ \mu_\mathbf{Y} \end{bmatrix}, \begin{bmatrix} \mathbf{\Sigma_X} & \mathbf{\Sigma_{XY}} \\ \mathbf{\Sigma_{YX}} & \mathbf{\Sigma_Y} \end{bmatrix} \right).$$

Here, $\mu_\mathbf{X} \in \mathbb{R}^{D_x \times 1}$ and $\mu_\mathbf{Y} \in \mathbb{R}^{D_y \times 1}$ are the means of data matrices $\mathbf{X}$ and $\mathbf{Y}$, respectively; covariance matrices $\mathbf{\Sigma_X}, \mathbf{\Sigma_Y}$ and the cross-covariance matrix $\mathbf{\Sigma_{XY}}$ are constructed as follows. Given ground truth projection matrices $\mathbf{U} \in \mathbb{R}^{D_x \times R}, \mathbf{V} \in \mathbb{R}^{D_y \times R}$ and canonical correlations $\boldsymbol{\rho} = (\rho_1, \rho_2, \ldots, \rho_R)$ defined in (13a). Let $\mathbf{U} = \mathbf{Q_X R_X}$ and $\mathbf{V} = \mathbf{Q_Y R_Y}$ be the QR decomposition of $\mathbf{U}$ and $\mathbf{V}$, then we have

$$\mathbf{\Sigma_{XY}} = \mathbf{\Sigma_X} \mathbf{U} \, \text{diag}(\boldsymbol{\rho}) \, \mathbf{V}^\top \mathbf{\Sigma_Y}, \tag{14a}$$

$$\mathbf{\Sigma_X} = \mathbf{Q_X R_X}^{-\top} \mathbf{R_X}^{-1} \mathbf{Q_X}^\top + \tau_x \mathbf{T_X} (\mathbf{I}_{D_x} - \mathbf{Q_X Q_X}^\top) \mathbf{T_X}^\top, \tag{14b}$$

$$\mathbf{\Sigma_Y} = \mathbf{Q_Y R_Y}^{-\top} \mathbf{R_Y}^{-1} \mathbf{Q_Y}^\top + \tau_y \mathbf{T_Y} (\mathbf{I}_{D_y} - \mathbf{Q_Y Q_Y}^\top) \mathbf{T_Y}^\top. \tag{14c}$$

Here, $\mathbf{T_X} \in \mathbb{R}^{D_x \times D_x}$ and $\mathbf{T_Y} \in \mathbb{R}^{D_y \times D_y}$ are randomly generated by normal distributions, and $\tau_x = 1$ and $\tau_y = 0.001$ are scaling hyperparameters. For subgroup distinction, we added noise to canonical vectors and adjusted sample sizes: 300, 350, 400, 450, and 500 observations each. In the numerical experiment, different canonical correlations are assigned to each subgroup alongside two global canonical vectors $\mathbf{U}$ and $\mathbf{V}$ to generate five distinct subgroups.

### 4.2.2 Real Data

**National Health and Nutrition Examination Survey (NHANES).** We utilized the 2005-2006 subset of the NHANES database https://www.cdc.gov/nchs/nhanes, including physical measurements and self-reported questionnaires from participants. We partitioned the data into two distinct subsets: one with 96 phenotypic measures and the other with 55 environmental measures. Our objective was to apply F-CCA to explore the interplay between phenotypic and environmental factors in contributing to health outcomes, considering the impact of education. Thus, we segmented the dataset into three subgroups based on educational attainment (i.e., lower than high school, high school, higher than high school), with 2,495, 2,203, and 4,145 observations in each subgroup.

Table 1: Numerical results in terms of Correlation ($\rho_r$), Maximum Disparity ($\Delta_{\max,r}$), and Aggregate Disparity ($\Delta_{\text{sum},r}$) metrics. Best values are in bold, and second-best are underlined. We focus on the initial five projection dimensions, but present only two dimensions here; results for other dimensions are in the supplementary material. We put the results of other projection dimensions in the supplementary material. "↑" means the larger the better and "↓" means the smaller the better. Note that MHAAPS has only 3 features, so we report results for its 1 and 2 dimensions.

| Dataset | Dim. (r) | $\rho_r \uparrow$ CCA | MF-CCA | SF-CCA | $\Delta_{\max,r} \downarrow$ CCA | MF-CCA | SF-CCA | $\Delta_{\text{sum},r} \downarrow$ CCA | MF-CCA | SF-CCA |
|---|---|---|---|---|---|---|---|---|---|---|
| Synthetic Data | 2 | **0.7533** | 0.7475 | 0.7309 | 0.3555 | 0.2866 | **0.2241** | 3.3802 | 2.8119 | **2.2722** |
| | 5 | **0.4717** | 0.4681 | 0.4581 | 0.4385 | 0.3313 | **0.2424** | 4.1649 | 3.1628 | **2.2304** |
| NHANES | 2 | **0.6392** | 0.6360 | 0.6334 | 0.0485 | 0.0359 | **0.0245** | 0.1941 | 0.1435 | **0.0980** |
| | 5 | **0.4416** | 0.4393 | 0.4392 | 0.1001 | **0.0818** | 0.0824 | 0.4003 | **0.3272** | 0.3297 |
| MHAAPS | 1 | **0.4464** | 0.4451 | 0.4455 | 0.0093 | 0.0076 | **0.0044** | 0.0187 | 0.0152 | **0.0088** |
| | 2 | **0.1534** | 0.1529 | 0.1526 | 0.0061 | 0.0038 | **0.0019** | 0.0122 | 0.0075 | **0.0039** |
| ADNI | 2 | **0.7778** | 0.7776 | 0.7753 | 0.0131 | 0.0119 | **0.0064** | 0.0263 | 0.0238 | **0.0127** |
| | 5 | **0.6810** | 0.6798 | 0.6770 | 0.0477 | 0.0399 | **0.0324** | 0.0954 | 0.0799 | **0.0648** |

Table 2: Mean computation time in seconds (±std) of 10 repeated experiments for $R = 5$ on the real dataset and $R = 7$ on the synthetic dataset. Experiments are run on Intel(R) Xeon(R) CPU E5-2660.

| Dataset | CCA | MF-CCA | SF-CCA |
|---|---|---|---|
| Synthetic Data | 0.0239±0.0026 | 109.0693±5.5418 | 29.1387±2.0828 |
| NHANES | 0.0483±0.0059 | 42.3186±1.9045 | 14.9156±1.8941 |
| MHAAPS | 0.0021±0.0047 | 3.5235±2.0945 | 0.8238±0.8155 |
| ADNI | 0.0039±0.0032 | 2.7297±0.5136 | 1.8489±1.0519 |

**Mental Health and Academic Performance Survey (MHAAPS).** This dataset is available at https://github.com/marks/convert_to_csv/tree/master/sample_data. It consists of three psychological variables, four academic variables, as well as sex information for a cohort of 600 college freshmen (327 females and 273 males). The primary objective of this investigation revolves around examining the interrelationship between the psychological variables and academic indicators, with careful consideration given to the potential influence exerted by sex.

**Alzheimer's Disease Neuroimaging Initiative (ADNI).** We utilized AV45 (amyloid) and AV1451 (tau) positron emission tomography (PET) data from the ADNI database (http://adni.loni.usc.edu) [73, 74]. ADNI data are analyzed for fairness in medical imaging classification [41, 53, 81], and sex disparities in ADNI's CCA study can harm generalizability, validity, and intervention tailoring. We utilized F-CCA to account for sex differences. Our experiment links 52 AV45 and 52 AV1451 features in 496 subjects (255 females, 241 males).

### 4.3 Results and Discussion

In the simulation experiment, we follow the methodology described in Section 4.2.1 to generate two sets of variables, each containing two subgroups of equal size. Canonical weights are trained and used to project the two sets of variables into a 2-dimensional space using CCA, SF-CCA, and MF-CCA. From Figure 2, it is clear that the angle between the distributions of the two subgroups, as projected by SF-CCA and MF-CCA, is smaller in comparison. This result indicates that F-CCA has the ability to reduce the disparity between distinct subgroups.

Table 1 shows the quantitative performance of the three models: CCA, MF-CCA, and SF-CCA. They are evaluated based on $\rho_r$, $\Delta_{\max,r}$, and $\Delta_{\text{sum},r}$ defined in (13) across five experimental sets. Table 2 displays the mean runtime of each model. Several key observations emerge from the analysis. Firstly, MF-CCA and SF-CCA demonstrate substantial improvements in fairness compared to CCA. However, it is important to note that F-CCA, employed in both MF-CCA and SF-CCA, compromises some degree of correlation due to its focus on fairness considerations during computations. Secondly,

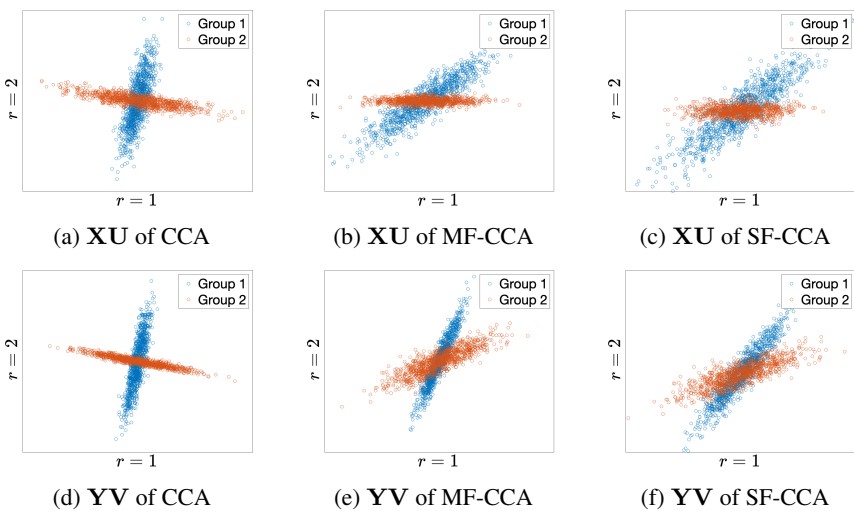

Figure 2: Scatter plot of the synthetic data points after projected to the 2-dimensional space. The distributions of the two groups after projection by CCA are orthogonal to each other. Our SF-CCA and MF-CCA can make the distributions of the two groups close to each other.

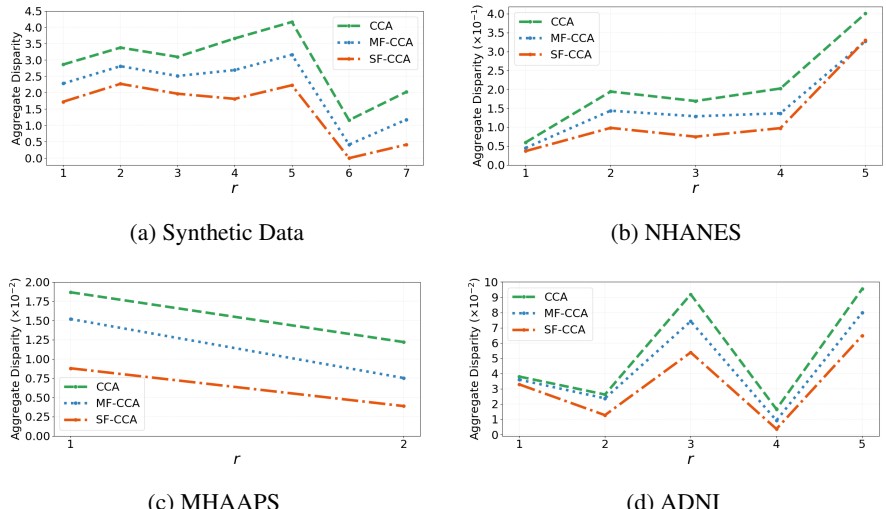

Figure 3: Aggregate disparity of CCA, MF-CCA, and SF-CCA (results from Table 1).

SF-CCA outperforms MF-CCA in terms of fairness improvement, although it sacrifices correlation. This highlights the effectiveness of the single-objective optimization approach in SF-CCA. Moreover, the datasets consist of varying subgroup quantities (5, 3, 2, and 2) and an imbalanced number of samples in distinct subgroups. F-CCA consistently performs well across these datasets, confirming its inherent scalability. Lastly, although SF-CCA requires more effort to tune hyperparameters, SF-CCA still exhibits a notable advantage in terms of time complexity compared to MF-CCA, demonstrating computational efficiency. Disparities among various CCA methods are visually represented in Figure 3. Notably, the conventional CCA consistently demonstrates the highest disparity error. Conversely, SF-CCA and MF-CCA consistently outperform CCA across all datasets, underscoring their efficacy in promoting fairness within analytical frameworks.

In Table 1, we define the *percentage change* of correlation ($\rho_r$), maximum disparity gap ($\Delta_{\max,r}$), and aggregate disparity ($\Delta_{\text{sum},r}$), respectively, as follows: $P\rho_r := (\rho_r \text{ of F-CCA} - \rho_r \text{ of CCA})/(\rho_r \text{ of CCA}) \times 100$, $P\Delta_{\max,r} := -(\Delta_{\max,r} \text{ of F-CCA} - \Delta_{\max,r} \text{ of CCA})/(\Delta_{\max,r} \text{ of CCA}) \times 100$, and $P\Delta_{\text{sum},r} := -(\Delta_{\text{sum},r} \text{ of F-CCA} - \Delta_{\text{sum},r} \text{ of CCA})/(\Delta_{\text{sum},r} \text{ of CCA}) \times 100$. Here, F-CCA is replaced with either MF-CCA or

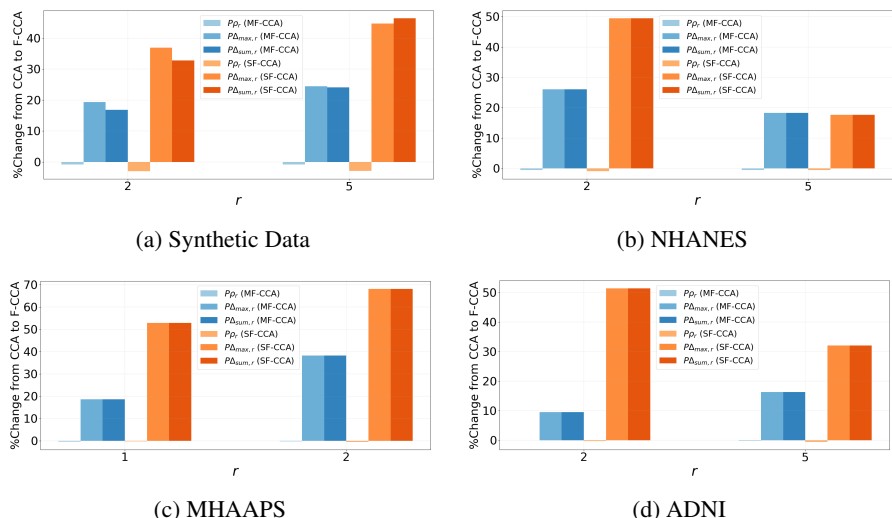

| (a) Synthetic Data | (b) NHANES |
|---|---|
| (c) MHAAPS | (d) ADNI |

Figure 4: Percentage change from CCA to F-CCA (results from Table 1). Each dataset panel shows two cases with projection dimensions ($r$). $P\rho_r$ is slight, while $P\Delta_{\max,r}$ and $P\Delta_{\text{sum},r}$ changes are substantial, signifying fairness improvement without significant accuracy sacrifice.

SF-CCA to obtain the percentage change for MF-CCA or SF-CCA. Figure 4 illustrates the percentage changes of each dataset. $P\rho_r$ is slight, while $P\Delta_{\max,r}$ and $P\Delta_{\text{sum},r}$ changes are substantial, signifying fairness improvement without significant accuracy sacrifice.

## 5 Conclusion, Limitations, and Future Directions

We propose F-CCA, a novel framework to mitigate unfairness in CCA. F-CCA aims to rectify the bias of CCA by learning global projection matrices from the entire dataset, concurrently guaranteeing that these matrices generate correlation levels akin to group-specific projection matrices. Experiments show that F-CCA is effective in reducing correlation disparity error without sacrificing much correlation. We discuss potential extensions and future problems stemming from our work.

- While F-CCA effectively reduces unfairness while maintaining CCA model accuracy, its potential to achieve a minimum achievable disparity correlation remains unexplored. A theoretical exploration of this aspect could provide valuable insights.

- F-CCA holds promise for extensions to diverse domains, including multiple modalities [80], deep CCA [3], tensor CCA [44], and sparse CCA [25]. However, these extensions necessitate novel formulations and in-depth analysis.

- Our approach of multi-objective optimization on smooth manifolds may find relevance in other problems, such as fair PCA [55]. Further, bilevel optimization approaches [37, 68, 65] can be designed on a smooth manifold to learn a single Pareto-efficient solution and provide an automatic trade-off between accuracy and fairness.

- With applications encompassing clustering, classification, and manifold learning, F-CCA ensures fairness when employing CCA techniques for these downstream tasks. It can also be jointly analyzed with fair clustering [15, 66, 34] and fair classification [78, 18].

## 6 Acknowledgements

This work was supported in part by the NIH grants U01 AG066833, U01 AG068057, RF1 AG063481, R01 LM013463, P30 AG073105, and U01 CA274576, and the NSF grant IIS 1837964. The ADNI data were obtained from the Alzheimer's Disease Neuroimaging Initiative database (https://adni.loni.usc.edu), funded by NIH U01 AG024904. Moreover, the NHANES data were sourced from the NHANES database (https://www.cdc.gov/nchs/nhanes).

We appreciate the reviewers' valuable feedback, which significantly improved this paper.

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

## Contents

# A    Addendum to Section 3

## A.1    Preliminaries on Retractions

In reference to Definition 1, the following options together with generalized polar decomposition defined in (4) represent commonly employed retractions[1, 6, 7, 21, 23, 38, 75] of a matrix $\boldsymbol{\xi} \in \mathbb{R}^{D \times R}$:

- **Exponential mapping.** It takes $8DR^2 + \mathcal{O}(R^3)$ flops and has the closed-form expression:

$$R_{\exp}^{\mathbf{z}}(\boldsymbol{\xi}) = [\mathbf{Z} \quad \mathbf{Q}] \exp \left( \begin{bmatrix} -\mathbf{Z}^\top \boldsymbol{\xi} & -\mathbf{R}^\top \\ \mathbf{R} & 0 \end{bmatrix} \right) \begin{bmatrix} \mathbf{I}_R \\ 0 \end{bmatrix},$$

  where $\mathbf{QR} = -(\mathbf{I}_r - \mathbf{ZZ}^\top)\boldsymbol{\xi}$ is the unique QR factorization.

- **Polar decomposition.** It takes $3DR^2 + \mathcal{O}(R^3)$ flops and has the closed-form expression:

$$R_{\text{polar}}^{\mathbf{z}}(\boldsymbol{\xi}) = (\mathbf{Z} + \boldsymbol{\xi})(\mathbf{I}_R + \boldsymbol{\xi}^\top \boldsymbol{\xi})^{-1/2}.$$

- **QR decomposition.** It takes $2DR^2 + \mathcal{O}(R^3)$ flops and has the closed-form expression:

$$R_{\text{qr}}^{\mathbf{z}}(\boldsymbol{\xi}) = \text{qr}(\mathbf{Z} + \boldsymbol{\xi}),$$

  where $\text{qr}(\mathbf{A})$ is the Q factor of the QR factorization of $\mathbf{A}$.

- **Cayley transformation.** It takes $7DR^2 + \mathcal{O}(R^3)$ flops and has the closed-form expression:

$$R_{\text{cayley}}^{\mathbf{Z}}(\boldsymbol{\xi}) = \left( \mathbf{I}_R - \frac{1}{2}\mathbf{W}(\boldsymbol{\xi}) \right)^{-1} \left( \mathbf{I}_R + \frac{1}{2}\mathbf{W}(\boldsymbol{\xi}) \right) \mathbf{Z},$$

  where $\mathbf{W}(\boldsymbol{\xi}) = (\mathbf{I}_R - \mathbf{ZZ}^\top/2)\boldsymbol{\xi}\mathbf{Z}^\top - \mathbf{Z}\boldsymbol{\xi}^\top(\mathbf{I}_R - \mathbf{ZZ}^\top/2)$.

This work specifically focuses on generalized polar decomposition defined in (4) for retraction.

## A.2    Subproblem Solver for MF-CCA

To solve the optimization problem (9), we introduce a method for finding the descent direction on the joint manifold, as described in Equation (10). In this section, we will provide a detailed explanation of the relevant calculations. We expand the optimization problem into two sub-optimization problems, one for updating the canonical weight $\mathbf{U}$ and the other for updating the canonical weight $\mathbf{V}$. Here, we focus on outlining the process of updating $\mathbf{U}$, which is described below.

In the $t$-th iteration, the gradient of the multi-objective function (9) with respect to $\mathbf{U}_t$ is computed as follows:

$$\nabla_{\mathbf{U}} f_1(\mathbf{U}_t, \mathbf{V}_t) = \frac{\partial f_1(\mathbf{U}_t, \mathbf{V}_t)}{\partial \mathbf{U}_t} = -\mathbf{X}^\top \mathbf{Y} \mathbf{V}_t,$$

$$\nabla_{\mathbf{U}} f_j(\mathbf{U}_t, \mathbf{V}_t) = \frac{\partial \Delta^{k,s}(\mathbf{U}_t, \mathbf{V}_t)}{\partial \mathbf{U}_t} \tag{15a}$$

$$= \phi' \left( \mathcal{E}^k(\mathbf{U}, \mathbf{V}) - \mathcal{E}^s(\mathbf{U}, \mathbf{V}) \right) \cdot \left( \frac{\partial \mathcal{E}^k(\mathbf{U}_t, \mathbf{V}_t)}{\partial \mathbf{U}_t} - \frac{\partial \mathcal{E}^s(\mathbf{U}_t, \mathbf{V}_t)}{\partial \mathbf{U}_t} \right),$$

for all $M \geq j \geq 2, k, s \in [K], k \neq s$, where

$$\frac{\partial \mathcal{E}^k(\mathbf{U}_t, \mathbf{V}_t)}{\partial \mathbf{U}_t} = \text{trace} \left( \mathbf{U}_t^{k,\star^\top} \mathbf{X}^{k^\top} \mathbf{Y}^k \mathbf{V}_t^{k,\star} \right) - \mathbf{X}^{k^\top} \mathbf{Y}^k \mathbf{V}_t,$$

$$\frac{\partial \mathcal{E}^s(\mathbf{U}_t, \mathbf{V}_t)}{\partial \mathbf{U}_t} = \text{trace} \left( \mathbf{U}_t^{s,\star^\top} \mathbf{X}^{s^\top} \mathbf{Y}^s \mathbf{V}_t^{s,\star} \right) - \mathbf{X}^{s^\top} \mathbf{Y}^s \mathbf{V}_t. \tag{15b}$$

Similarly, we can compute the gradient of $f$ w.r.t $\mathbf{V}_t$.

Lemma 7, inspired by [23, 6], states that the direction $\mathbf{P}_t$ can be expressed as a linear combination of gradients $\{\nabla f_1(\mathbf{U}_t, \mathbf{V}_t), \nabla f_2(\mathbf{U}_t, \mathbf{V}_t), \ldots, \nabla f_M(\mathbf{U}_t, \mathbf{V}_t)\}$.

**Lemma 7.** *The unconstrained optimization problem in* (10) *has a unique solution. Moreover,* $\mathbf{P}_t$ *is the solution of the problem in* (10) *if only if there exist* $\mu_i \geq 0, i \in [M]$, *such that*

$$\mathbf{P}_t = - \sum_{i \in [M]} \mu_i \nabla f_i(\mathbf{U}_t, \mathbf{V}_t), \quad \sum_{i \in [M]} \mu_i = 1, \quad \mu_i \geq 0, \text{ for } 1 \leq i \leq M. \tag{16}$$

*Proof.* Recall that for each $(\mathbf{U}, \mathbf{V}) \in \mathcal{U} \times \mathcal{V}$, we have $\mathbf{P} = (\mathbf{P}^{\mathbf{u}}, \mathbf{P}^{\mathbf{v}})$ with $\mathbf{P}^{\mathbf{u}} \in \mathcal{T}_{\mathbf{U}}\mathcal{U}$ and $\mathbf{P}^{\mathbf{v}} \in \mathcal{T}_{\mathbf{V}}\mathcal{V}$. It can be easily seen that the first term of $Q(\mathbf{P})$ (defined in (10)) is a summation of the maximum of linear functions in the linear space $\mathcal{T}_{\mathbf{U}}\mathcal{U} \times \mathcal{T}_{\mathbf{V}}\mathcal{V}$. Hence, it is convex, which implies that $Q(\mathbf{P})$ is strongly convex. Thus, the solution is unique.

From the convexity of the function, it is well known that $\mathbf{P}_t$ is the solution of (10) if only if

$$\mathbf{0} \in \partial Q(\mathbf{P})\mathbf{P}_t,$$

or equivalently,

$$-\mathbf{P}_t \in \partial \left( \max_{i \in [M]} \operatorname{trace} \left( \mathbf{P}^{\top} \nabla f_i(\mathbf{U}_t, \mathbf{V}_t) \right) \right) \mathbf{P}_t.$$

Therefore, the second statement follows from the formula for the subdifferential of the maximum of convex functions. $\square$

The (sub)-gradient formula (15) can be substituted into Equation (10), and subsequently, the *fminimax* function from the MATLAB optimization toolbox can be applied to solve for the descent direction $\mathbf{P}_t$. We found that the Goal Attainment Method [24] gives a more stable and accurate solution for our subproblem [3]. Given $\mathbf{P}_t^{\mathbf{u}}$, $\mathbf{U}_t$ can be updated by:

$$\mathbf{U}_{t+1} = R^{\mathbf{u}}(\eta_t \mathbf{P}_t^{\mathbf{u}}), \tag{17}$$

where $\eta_t$ is the adaptive stepsize and $R^{\mathbf{u}}$ is the retraction operator onto $\mathcal{U} = \{\mathbf{U} \in \mathbb{R}^{D_x \times R} | \mathbf{U}^{\top}\mathbf{X}^{\top}\mathbf{X}\mathbf{U} = \mathbf{I}_R\}$.

Various retraction options are discussed in Sections 3.1 and A.1. Specifically, the generalized polar decomposition method defined in (4) is employed in our numerical experiments described in Section 4. The symmetric nature of the update process for canonical weights $\mathbf{U}$ and $\mathbf{V}$ in CCA allows us to employ identical procedures for updating $\mathbf{V}$. By substituting $\mathbf{X}$ with $\mathbf{Y}$ and $\mathbf{U}$ with $\mathbf{V}$ in the aforementioned steps, we obtain the process for updating $\mathbf{V}$. In each iteration, $\mathbf{U}$ and $\mathbf{V}$ are updated once in sequence.

### A.3 Subproblem Solver for SF-CCA

To solve the optimization problem (11), the method of finding the descent direction is introduced in Equation (12). Detailed calculations will be explained below. We expand it into two sub-optimization problems to update the canonical weight $\mathbf{U}$ and the canonical weight $\mathbf{V}$, respectively. First, to update the canonical weight $\mathbf{U}_t$ in iteration $t$, we can take gradient of $f(\mathbf{U}_t, \mathbf{V}_t)$ with respect $\mathbf{U}_t$ as follows:

$$
\begin{aligned}
\nabla f(\mathbf{U}_t, \mathbf{V}_t) &= \frac{\partial f(\mathbf{U}_t, \mathbf{V}_t)}{\partial \mathbf{U}_t} \\
&= -\mathbf{X}^{\top}\mathbf{Y}\mathbf{V}_t + \lambda \frac{\partial \Delta(\mathbf{U}_t, \mathbf{V}_t)}{\partial \mathbf{U}_t} \\
&= -\mathbf{X}^{\top}\mathbf{Y}\mathbf{V}_t \\
&\quad + \lambda \left( \sum_{k,s \in [K], k \neq s} \phi' \left( \mathcal{E}^k(\mathbf{U}, \mathbf{V}) - \mathcal{E}^s(\mathbf{U}, \mathbf{V}) \right) \cdot \left( \frac{\partial \mathcal{E}^k(\mathbf{U}_t, \mathbf{V}_t)}{\partial \mathbf{U}_t} - \frac{\partial \mathcal{E}^s(\mathbf{U}_t, \mathbf{V}_t)}{\partial \mathbf{U}_t} \right) \right),
\end{aligned} \tag{18a}
$$

where

$$
\begin{aligned}
\frac{\partial \mathcal{E}^k(\mathbf{U}_t, \mathbf{V}_t)}{\partial \mathbf{U}_t} &= \operatorname{trace} \left( \mathbf{U}_t^{k,\star^{\top}} \mathbf{X}^{k^{\top}} \mathbf{Y}^k \mathbf{V}_t^{k,\star} \right) - \mathbf{X}^{k^{\top}} \mathbf{Y}^k \mathbf{V}_t, \\
\frac{\partial \mathcal{E}^s(\mathbf{U}_t, \mathbf{V}_t)}{\partial \mathbf{U}_t} &= \operatorname{trace} \left( \mathbf{U}_t^{s,\star^{\top}} \mathbf{X}^{s^{\top}} \mathbf{Y}^s \mathbf{V}_t^{s,\star} \right) - \mathbf{X}^{s^{\top}} \mathbf{Y}^s \mathbf{V}_t.
\end{aligned} \tag{18b}
$$

---

[3] https://ww2.mathworks.cn/help/optim/ug/multiobjective-optimization-algorithms.html

Given the subgradients (18) and the single-objective subproblem (12), the direction $\mathbf{G}_t = (\mathbf{G}_t^{\mathbf{u}}, \mathbf{G}_t^{\mathbf{v}})$ can be effectively computed using subgradient-based methods such as the *fmincon* function from the MATLAB optimization toolbox. Given $\mathbf{G}_t^{\mathbf{u}}$, the update for $\mathbf{U}_t$ can be obtained as follows:

$$\mathbf{U}_{t+1} = R^{\mathbf{u}}(\eta_t \mathbf{G}_t^{\mathbf{u}}). \tag{19}$$

Here, $\eta_t$ is the adaptive stepsize and $R^{\mathbf{u}}$ is the retraction operator onto $\mathcal{U} = \{\mathbf{U} \in \mathbb{R}^{D_x \times R} | \mathbf{U}^{\top}\mathbf{X}^{\top}\mathbf{X}\mathbf{U} = \mathbf{I}_R\}$.

In our numerical experiments, we utilize polar decomposition as the retraction operator; see (4). Similarly, the canonical weight $\mathbf{V}$ can be updated using the same procedure.

## A.4   Auxiliary Lemmas

**Lemma 8.** *Suppose Assumption A holds. For any $(\mathbf{U}_t, \mathbf{V}_t) \in \mathcal{U} \times \mathcal{V}$, let $\mathbf{P}_t$ be the solution of (10). Then, for any $\eta_t \geq 0$, we have the following:*

$$\mathbf{F}(\mathbf{U}_{t+1}, \mathbf{V}_{t+1}) \preceq \mathbf{F}(\mathbf{U}_t, \mathbf{V}_t) + \eta_t \boldsymbol{\nabla}_t + \eta_t^2 \frac{L_F}{2} \|\mathbf{P}_t\|_{\mathrm{F}}^2 \mathbf{1}_M, \qquad \mathbf{P}_t \in \mathcal{T}_{\mathbf{U}}\mathcal{U} \times \mathcal{T}_{\mathbf{V}}\mathcal{V},$$

*where $\nabla_{it} := \langle \nabla f_i(\mathbf{U}_t, \mathbf{V}_t), \mathbf{P}_t \rangle$ and $\boldsymbol{\nabla}_t := [\nabla_{1t}, \cdots, \nabla_{Mt}]^{\top} \in \mathbb{R}^M$.*

*Proof.* The proof is a straightforward consequence of Assumption A. $\qquad\square$

**Lemma 9.** *For any $(\mathbf{U}_t, \mathbf{V}_t) \in \mathcal{U} \times \mathcal{V}$, let $\mathbf{P}_t$ be the solution of (10). Then,*

$$\max_{i \in [M]} \operatorname{trace}\left(\mathbf{P}_t^{\top} \nabla f_i(\mathbf{U}_t, \mathbf{V}_t)\right) = -\|\mathbf{P}_t\|_{\mathrm{F}}^2. \tag{20}$$

*Hence,* $\operatorname{trace}\left(\mathbf{P}_t^{\top} \nabla f_i(\mathbf{U}_t, \mathbf{V}_t)\right) \preceq -\|\mathbf{P}_t\|_{\mathrm{F}}^2$. *Further, $\mathbf{P}_t$ is critical Pareto point of $\mathbf{F}$ if, and only if, $\|\mathbf{P}_t\|_{\mathrm{F}} = 0$.*

*Proof.* It follows from (10) that

$$-\|\mathbf{P}_t\|_{\mathrm{F}}^2 = \operatorname{trace}\left(\mathbf{P}_t^{\top} \sum_{i \in [M]} \mu_i \nabla f_i(\mathbf{U}_t, \mathbf{V}_t)\right) = \sum_{i \in [M]} \mu_i \operatorname{trace}\left(\mathbf{P}_t^{\top} \nabla f_i(\mathbf{U}_t, \mathbf{V}_t)\right).$$

Hence, by the second equality in (16), it is easy to verify that (20) holds. The second statement follows by using the definitions of $\operatorname{trace}(\mathbf{P}_t^{\top} \nabla F((\mathbf{U}_t, \mathbf{V}_t))$. We proceed with the proof of the third statement of the lemma. Assuming that $\mathbf{P}_t$ is a critical Pareto, it follows from the definition that there exists $i \in [M]$ such that $\operatorname{trace}(\mathbf{P}_t^{\top} \nabla f_i(\mathbf{U}_t, \mathbf{V}_t) \geq 0$. Then, by the first part of the lemma, we have $\|\mathbf{P}_t\| = 0$. The converse (only if) follows from the application of [6, Lemma 4.2]. $\qquad\square$

The following result provides an estimate for the decrease of a function $\mathbf{F}$, along $\mathbf{P}$. This result is crucial in determining the iteration-complexity bounds for the gradient method on a general Riemannian manifold.

**Lemma 10** (Descent Property of MF-CCA). *Suppose Assumption A holds. For any $(\mathbf{U}_t, \mathbf{V}_t) \in \mathcal{U} \times \mathcal{V}$, let $\mathbf{P}_t$ be the solution of (10). Then, for any $\eta_t \geq 0$, we have*

$$\mathbf{F}(\mathbf{U}_{t+1}, \mathbf{V}_{t+1}) \preceq \mathbf{F}(\mathbf{U}_t, \mathbf{V}_t) + \left(\frac{L_F \eta_t^2}{2} - \eta_t\right) \|\mathbf{P}_t\|_{\mathrm{F}}^2 \mathbf{1}_M.$$

*Proof.* The proof is a straight combination of Lemmas 8 and 9. $\qquad\square$

## A.5   Proof of Theorem 5

*Proof.* It follows from Lemma 10 that

$$\left(\frac{L_F \eta_t^2}{2} - \eta_t\right) \|\mathbf{P}_t\|_{\mathrm{F}}^2 \mathbf{1}_M \preceq \mathbf{F}(\mathbf{U}_t, \mathbf{V}_t) - \mathbf{F}(\mathbf{U}_{t+1}, \mathbf{V}_{t+1}),$$

for all $t = 0, 1, \ldots, T - 1$.

Setting $\eta_t = \eta$, and summing both sides of this inequality for $t = 0, 1, \ldots, T-1$, we get

$$\left( \frac{L_F \eta^2}{2} - \eta \right) \sum_{t=0}^{T-1} \|\mathbf{P}_t\|_{\mathrm{F}}^2 \, \mathbf{1}_M \preceq \mathbf{F}(\mathbf{U}_0, \mathbf{V}_0) - \mathbf{F}(\mathbf{U}_T, \mathbf{V}_T).$$

Thus, by the definition of $i_*$, we obtain

$$\left( \frac{L_F \eta^2}{2} - \eta \right) T \min \left\{ \|\mathbf{P}_t\|_{\mathrm{F}}^2 : \ t = 0, 1, \ldots, T-1 \right\} \leq f_{i_*}(\mathbf{U}_0, \mathbf{V}_0) - f_{i_*}^*,$$

which together with $\eta \leq \frac{1}{L_F}$ implies

$$\min \left\{ \|\mathbf{P}_t\|_{\mathrm{F}} : \ t = 0, \ldots, T-1 \right\} \leq \frac{2}{\eta} \left[ \frac{f_{i_*}(\mathbf{U}_0, \mathbf{V}_0) - f_{i_*}^*}{T} \right]^{\frac{1}{2}}.$$

$\square$

**Remark 11.** *As a consequence, if we consider a tolerance level of $\epsilon$, we can bound the number of iterations required by the gradient method to obtain $\mathbf{P}_T$ such that $\|\mathbf{P}_T\|_{\mathrm{F}} \leq \epsilon$. This bound can be expressed as $\mathcal{O}\left( (f_{i_*}(\mathbf{U}_0, \mathbf{V}_0) - f_{i_*}^*)/\epsilon^2 \right)$, where $f_{i_*}(\mathbf{U}_0, \mathbf{V}_0) - f_{i_*}^* = \min \left\{ f_i(\mathbf{U}_0, \mathbf{V}_0) - f_i^* : \ i \in [M] \right\}$. In other words, the number of iterations needed by the gradient method to achieve a solution $(\mathbf{U}_T, \mathbf{V}_T)$ with $\|\mathbf{P}_T\|_{\mathrm{F}} \leq \epsilon$ is proportional to the difference between the initial objective function value and the optimal objective function value, divided by the square of the tolerance level $\epsilon$. The "big O" notation, $\mathcal{O}$, indicates that the bound is an upper bound, providing an estimate of the worst-case behavior of the algorithm. This result showcases the convergence rate of the gradient method and provides a useful guideline for choosing the tolerance level and estimating the computational resources required to achieve the desired accuracy.*

### A.6 Proof of Theorem 6

*Proof.* This proof is a slightly modified version of the proof of Theorem 5. Using Assumption B and a single-objective variant of Lemma 10, we obtain

$$\begin{aligned} &f\left(\mathbf{U}_{t+1}, \mathbf{V}_{t+1}\right) - f\left(\mathbf{U}_t, \mathbf{V}_t\right) \\ &\leq \left\langle \nabla f\left(\mathbf{U}_t, \mathbf{V}_t\right), \left((\mathbf{U}_{t+1}, \mathbf{V}_{t+1}) - (\mathbf{U}_t, \mathbf{V}_t)\right) \right\rangle + \frac{L_f}{2} \left\| (\mathbf{U}_{t+1}, \mathbf{V}_{t+1}) - (\mathbf{U}_t, \mathbf{V}_t) \right\|_{\mathrm{F}}^2 \\ &= -\eta \left\| \nabla f\left(\mathbf{U}_t, \mathbf{V}_t\right) \right\|_{\mathrm{F}}^2 + \frac{L_f \eta^2}{2} \left\| \nabla f\left(\mathbf{U}_t, \mathbf{V}_t\right) \right\|_{\mathrm{F}}^2 \\ &= -\eta \left( 1 - \frac{L_f \eta}{2} \right) \left\| \nabla f\left(\mathbf{U}_t, \mathbf{V}_t\right) \right\|_{\mathrm{F}}^2. \end{aligned}$$

Summing both sides of this inequality for $t = 0, 1, \ldots, T-1$, we get

$$\left( \frac{L_f \eta^2}{2} - \eta \right) \sum_{t=0}^{T-1} \|\mathbf{G}_t\|_{\mathrm{F}}^2 \leq f(\mathbf{U}_0, \mathbf{V}_0) - f(\mathbf{U}_T, \mathbf{V}_T).$$

Hence, using the fixed stepsize $\eta_t = \eta \leq \frac{1}{L_f}$, we get

$$\min \left\{ \|\mathbf{G}_t\|_{\mathrm{F}} : \ t = 0, \ldots, T-1 \right\} \leq \frac{2}{\eta} \left[ \frac{f(\mathbf{U}_0, \mathbf{V}_0) - f^*}{T} \right]^{\frac{1}{2}}.$$

$\square$

**Remark 12.** *As a result, given a tolerance $\epsilon$, the number of iterations required by the gradient method to obtain $(\mathbf{U}_T, \mathbf{V}_T)$ such that $\|\mathbf{G}_T\|_{\mathrm{F}} \leq \epsilon$ is bounded by $\mathcal{O}\left( (f(\mathbf{U}_0, \mathbf{V}_0) - f^*)/\epsilon^2 \right)$.*

Table 3: Statistic summary of real datasets. In the ADNI database, AV45 is used twice in two different datasets but with a different number of features. In AV45 vs. AV1451, we use the shared ROI (region of interest), thus, the two views have the same number of features. While in AV45 vs Cognitive Score, AV45 contains the total number of features.

| Datbase | Dataset | Number of Features | Number of Samples | Sensitive Attribute | Group Distribution |
|---|---|---|---|---|---|
| NHANES | Phenotypic Measures | 96 | 8843 | Education | <High School: 2495 = High School: 2203 >High School: 4145 |
|  | Environmental Measures | 55 |  |  |  |
| NHANES | Phenotypic Measures | 96 | 8136 | Race | White: 4576 Black: 1793 Mexican: 1767 |
|  | Environmental Measures | 55 |  |  |  |
| MHAAPS | Psychological Performance Academic Performance | 3 4 | 600 | Sex | Male: 273 Female: 327 |
| ADNI | AV45 AV1451 | 52 52 | 496 | Sex | Male: 241 Female: 255 |
| ADNI | AV45 Cognitive Score | 68 46 | 785 | Sex | Male: 431 Female: 354 |

## B  Addendum to Section 4

### B.1  Detailed Description of Datasets

Table 3 provides a comprehensive overview of the relevant details associated with each real dataset. These details include the number of samples, the number of features, sensitive attributes, and group distribution. Notably, within the group distribution, it is important to acknowledge that in certain datasets, the total number of samples within different sensitive attributes may not be equivalent. This discrepancy arises due to the presence of missing information, as well as the insufficient sample sizes in certain subgroups. Figure 5 visualizes the distribution of all the groups in each dataset. We can see that the groups are imbalanced in synthetic data, NHANES (Education) and NHANES (Race), while balanced in MHAAPS (Sex), ADNI AV45/AV1451 (Sex), and ADNI AV45/Cognition (Sex). Imbalanced groups can make the problem more complicated, which is a good challenge to validate the effectiveness of our methods.

In the following, we provide comprehensive descriptions of all the datasets we used.

#### B.1.1  National Health and Nutrition Examination Survey (NHANES)

The NHANES dataset is extensively used for CCA analysis [48] and fairness studies [17, 52]. The dataset's diverse attributes give rise to fairness considerations, underscoring the importance of conducting equitable CCA experiments to tackle discrepancies in sample representation stemming from these inequalities.

In our study, we narrow our focus to the 2005-2006 subset of the NHANES database available at https://www.cdc.gov/nchs/nhanes. This subset comprises physical measurements and self-reported questionnaires from participants. To address missing data concerns, we employ the Multiple Imputation by Chained Equations (MICE) Forest methodology. Afterward, we divide the data into two distinct subsets: the *"Phenotypic-Measure"* dataset and the *"Environmental-Measure"* dataset, based on the inherent nature of the features. This stratified approach enables us to leverage F-CCA to gain a nuanced understanding of how phenotypic and environmental factors interact in influencing health outcomes while also accounting for the potential influence of education and race.

In our numerical experiments, we utilize education and race as sensitive attributes to partition the two datasets. In the initial experiment, we split the dataset into three subgroups based on participants' educational attainment. These subgroups comprise 2495, 2203, and 4145 observations, respectively. For the subsequent experiment, we work with a total of 8136 observations, dividing them into three

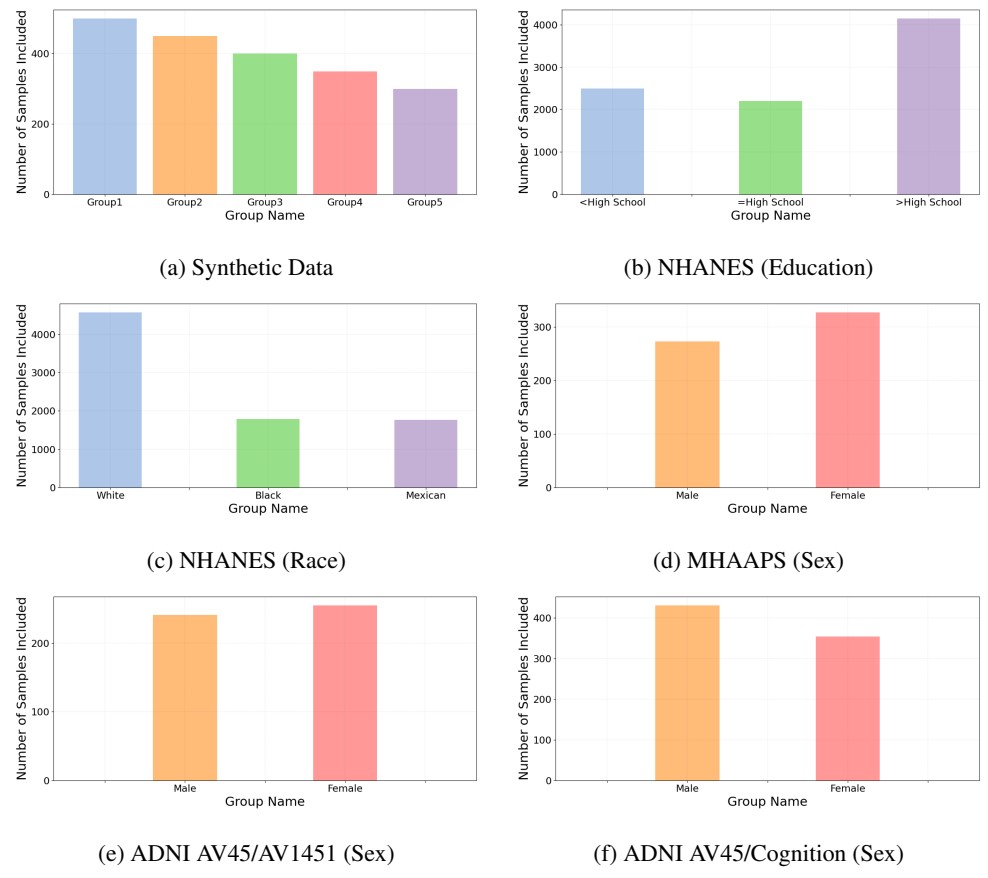

|  |  |
|---|---|
| (a) Synthetic Data | (b) NHANES (Education) |
| (c) NHANES (Race) | (d) MHAAPS (Sex) |
| (e) ADNI AV45/AV1451 (Sex) | (f) ADNI AV45/Cognition (Sex) |

Figure 5: Group distributions of the studied datasets.

subgroups based on racial categories. Specifically, there are 4576 white subjects, 1793 black subjects, and 1767 Mexican subjects. The *"Phenotypic-Measure"* dataset encompasses 96 distinct features, while the *"Environmental-Measure"* dataset includes 55 features.

### B.1.2 Mental Health and Academic Performance Survey (MHAAPS)

The dataset employed in this study is obtained from the online repository available at `https://github.com/marks/convert_to_csv/tree/master/sample_data`. This particular dataset includes three distinct psychological variables and four academic variables in the form of standardized test scores, as well as gender information for a cohort of 600 individuals classified as college freshmen. The primary objective of this investigation revolves around examining the interrelationship between the aforementioned psychological variables and academic indicators, with careful consideration given to the potential influence exerted by gender. Specifically, the dataset consists of 327 female samples and 273 male samples.

### B.1.3 Alzheimer's Disease Neuroimaging Initiative (ADNI)

We utilize AV45 (amyloid), AV1451 (tau) positron emission tomography (PET), and Cognitive Score data from the Alzheimer's Disease Neuroimaging Initiative (ADNI) database (`http://adni.loni.usc.edu`) as three of our experiment datasets. Over- or under-representation of age, sex, or biometric groups might affect ADNI data fairness. Disease distribution among sensitivity features, such as higher AD occurrence in women, can also impact fairness [41]. The unfairness in sex representation within the CCA study of the ADNI dataset arises from an imbalance in the number of male and female participants. This disparity undermines the generalizability and validity of the study findings, as it fails to account for potential sex-related differences in Alzheimer's disease progression and response to treatments. The skewed representation limits the ability to draw accurate conclusions

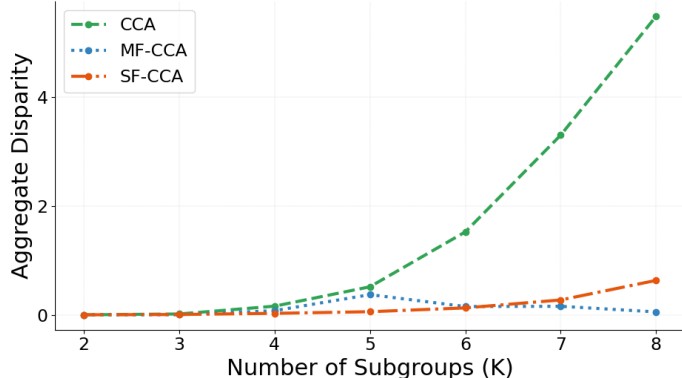

Figure 6: Aggregate disparity of 1st projection dimension on synthetic data comprising varying numbers of subgroups ($K$).

and develop tailored interventions for both sexes, perpetuating sex bias in research and healthcare. Consequently, it becomes imperative to utilize F-CCA as a means to acknowledge and integrate sex disparities within research initiatives.

In the conducted numerical study, two distinct experiments are carried out utilizing the ADNI dataset, where the sensitive attribute considered is the sex of the participants. The first experiment aims to explore the correlation between AV45 and AV1451, whereas the second experiment focuses on investigating the correlation between AV45 and Cognitive Scores. In the first experiment, both AV45 and AV1451 include 52 regional features and comprise a total of 496 observations. Among these observations, 255 belong to the female subgroup, and 241 belong to the male subgroup. In the analysis involving the correlation between AV45 and Cognitive Scores, a subset of 785 common samples is extracted, consisting of 431 male samples and 354 female samples. This analysis employs 68 regional features associated with AV45 and 46 cognitive score features.

## B.2 Trade-off Analysis of Correlation and Disparity

### B.2.1 Sensitivity of Correlation and Disparity Error to $K$

We conducted an analysis to examine the sensitivity of fairness in relation to the number of subgroups $K$. To investigate this, we utilized synthetic data consisting of varying numbers of subgroups and compared the resulting aggregate disparity on the first projection dimension.

Figure 6 provides a visual representation of our findings. The results clearly demonstrate that as the number of subgroups increases, there is a substantial amplification in the aggregate disparity when employing CCA. However, both MF-CCA and SF-CCA effectively address this issue and mitigate the observed phenomenon. Specifically, they successfully reduce the level of disparity even as the number of subgroups grows.

Moreover, it is worth noting that the aggregate discrepancy of SF-CCA exhibits a certain degree of sensitivity beyond a threshold of six subgroups. Beyond this point, the effectiveness of SF-CCA in reducing disparity begins to diminish. Conversely, MF-CCA remains consistently unaffected by variations in the number of subgroups, consistently maintaining its fairness performance. This analysis underscores the robustness and scalability of MF-CCA in handling an increasing number of subgroups, as it consistently maintains fairness irrespective of this parameter. On the other hand, SF-CCA demonstrates effective fairness mitigation but exhibits limitations when faced with a large number of subgroups beyond a certain threshold.

### B.2.2 Sensitivity of Correlation and Disparity Error to $r$

Table 4 presents the numerical results of three measures described in Section 4.1, namely correlation ($\rho_r$), maximum disparity ($\Delta_{\max,r}$), and aggregate disparity ($\Delta_{\text{sum},r}$). The results are with respect to

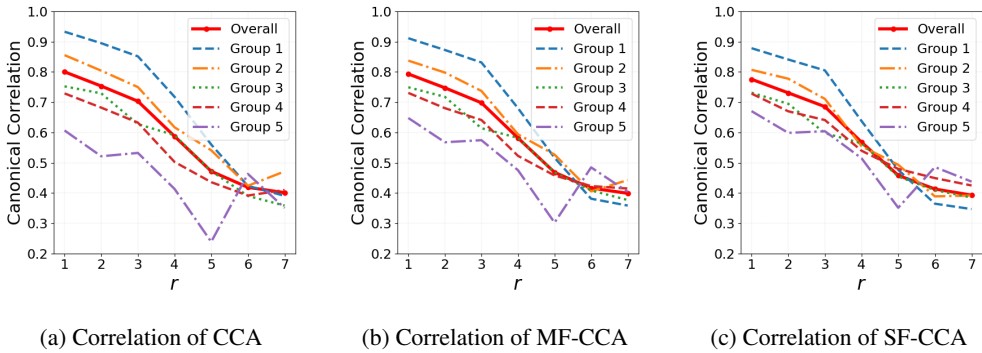

| (a) Correlation of CCA | (b) Correlation of MF-CCA | (c) Correlation of SF-CCA |
|---|---|---|

Figure 7: Visualization of the canonical correlation results on synthetic data for the total five projection dimensions ($r$). All the methods are applied to both the entire dataset and individual subgroups. The closer each subgroup's curve is to the overall curve, the better.

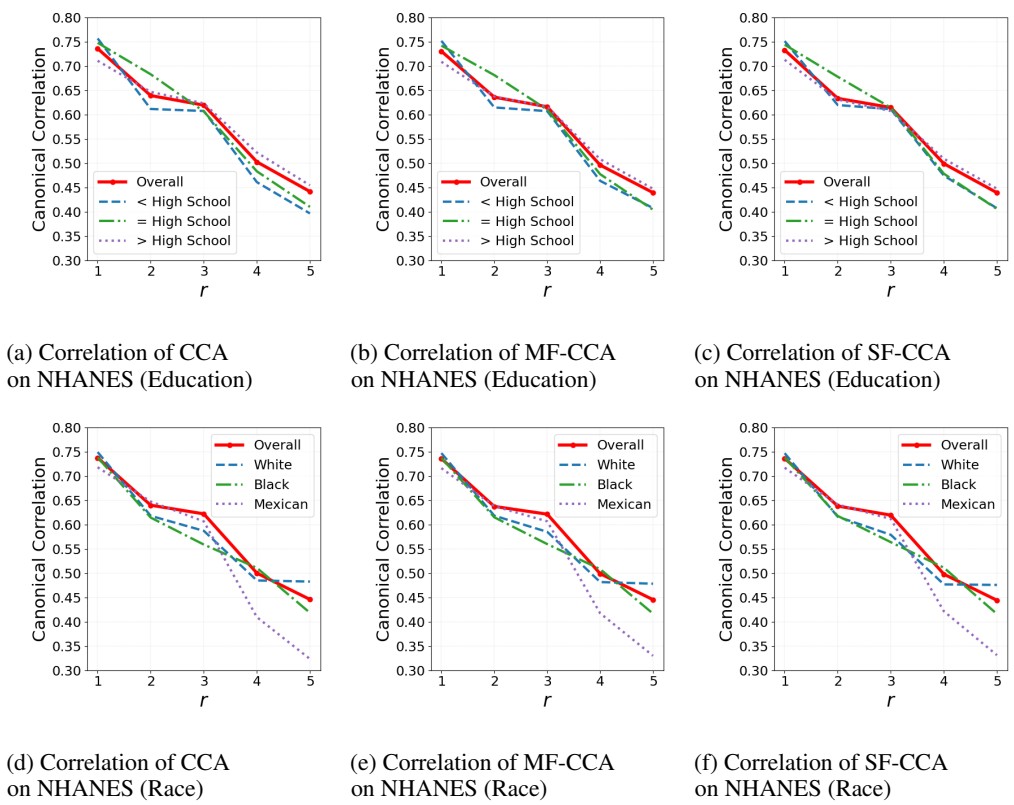

| (a) Correlation of CCA on NHANES (Education) | (b) Correlation of MF-CCA on NHANES (Education) | (c) Correlation of SF-CCA on NHANES (Education) |
|---|---|---|
| (d) Correlation of CCA on NHANES (Race) | (e) Correlation of MF-CCA on NHANES (Race) | (f) Correlation of SF-CCA on NHANES (Race) |

Figure 8: Visualization of the canonical correlation results of NHANES (Education & Race) for the total five projection dimensions ($r$). All the methods are applied to both the entire dataset and individual subgroups. The closer each subgroup's curve is to the overall curve, the better.

the first $r$ projection dimensions. It is an extension of Table 1. Specifically, we present the results of the first seven projection dimensions for synthetic data; the first five projection dimensions for NHANES and ADNI; and the first two projection dimensions for MHAAPS, respectively. Overall, we have consistent results and conclusions in Table 1. Firstly, MF-CCA and SF-CCA show substantial improvements in fairness compared to CCA with mild compromises of correlation. Secondly, SF-CCA outperforms MF-CCA in terms of fairness improvement, although it sacrifices correlation. This highlights the effectiveness of the single-objective optimization approach in SF-CCA. F-CCA consistently performs well across these datasets, confirming its inherent scalability.

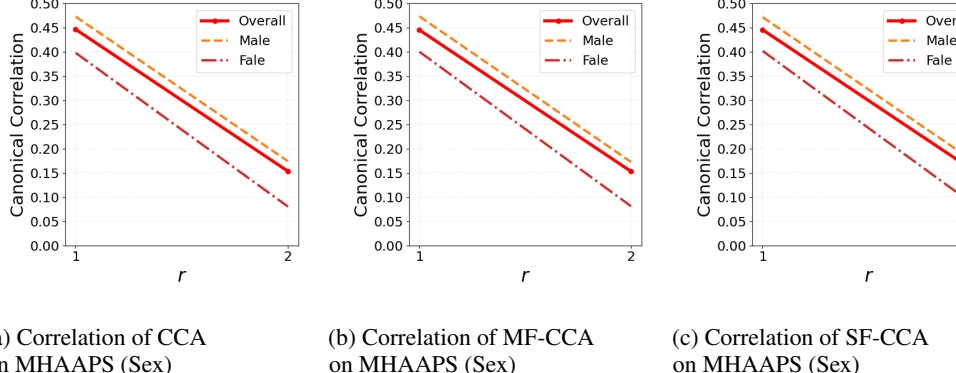

(a) Correlation of CCA on MHAAPS (Sex)

(b) Correlation of MF-CCA on MHAAPS (Sex)

(c) Correlation of SF-CCA on MHAAPS (Sex)

Figure 9: Visualization of the canonical correlation results of MHAAPS (Sex) for the total two projection dimensions ($r$). All the methods are applied to both the entire dataset and individual subgroups. The closer each subgroup's curve is to the overall curve, the better.

Table 5 quantifies the improvement in fairness and the minor compromises in accuracy presented in Table 4 by depicting the percentage performance shift from baseline CCA to MF-CCA and SF-CCA. The table is a reference and an extension of Figure 4 in Section 4.1. Figure 4 only visualizes the percentage shifts of the second and fifth projection dimension for Synthetic Data, NHANES, and ADNI and the first two projection dimensions for MHAAPS regarding the results presented in Table 1 for coherence. In Table 5, the metrics are formulated as: $P\rho_r = \frac{\rho_r \text{ of F-CCA} - \rho_r \text{ of CCA}}{\rho_r \text{ of CCA}} \times 100$, $P\Delta_{\max,r} = -\frac{\Delta_{\max,r} \text{ of F-CCA} - \Delta_{\max,r} \text{ of CCA}}{\Delta_{\max,r} \text{ of CCA}} \times 100$, $P\Delta_{\text{sum},r} = -\frac{\Delta_{\text{sum},r} \text{ of F-CCA} - \Delta_{\text{sum},r} \text{ of CCA}}{\Delta_{\text{sum},r} \text{ of CCA}} \times 100$, where F-CCA is switched to MF-CCA and SF-CCA to obtain the corresponding shift results. The formulations are designed regarding the properties of the metrics where $\rho_r$ is the larger the better while $\Delta_{\max,r}$ and $\Delta_{\text{sum},r}$ are the smaller the better. According to the table, compared to CCA, MF-CCA, and SF-CCA sacrifice slightly the performance in correlation ($\rho_r$) in exchange for significant fairness improvements in terms of maximum disparity ($\Delta_{\max,r}$) and aggregate disparity ($\Delta_{\text{sum},r}$) across all datasets in general, which demonstrate the good performance of F-CCA.

We also provide visualizations demonstrating how the correlation changes as the dimensionality changes. Figures 7, 8, 9, and Figure 10 display these correlation curves. Our expectation is that the correlation curves of our methods will show a stronger tendency to concentrate around the overall correlation curve compared to the baseline method. The findings indicate that the utilization of both MF-CCA and SF-CCA approaches yields a convergence of subgroup performance towards the overall canonical correlation. Furthermore, the overall canonical correlations derived from these two models exhibit a comparable level of performance to that of the CCA model.

Upon examining the synthetic data in Figure 7, we observe a clear concentration tendency. However, in the case of real data, this tendency is less pronounced. This is due to the fact that the disparities between different groups in the real data are already quite small. As a result, all the dashed curves in the CCA are already close to the solid red curve.

### B.2.3 Sensitivity of Correlation and Disparity of SF-CCA to $\lambda$

The SF-CCA experiment incorporates the hyperparameter $\lambda$ to strike a balance between correlation and fairness considerations. This hyperparameter allows us to modulate the emphasis placed on fairness within the SF-CCA model. Figure 11 provides a visual representation of the effects of different $\lambda$ values on both correlation and fairness metrics.

The illustration clearly demonstrates that increasing the magnitude of $\lambda$ enhances the emphasis on fairness within the SF-CCA model. Consequently, this leads to a reduction in aggregate disparity, indicating improved fairness. Conversely, lower values of $\lambda$ prioritize correlation, emphasizing the preservation of the underlying relationship between the datasets.

This experimental finding aligns with the discussion presented in Section 3 of our work, where we discussed the trade-off between correlation and fairness. By adjusting the value of $\lambda$, we can effectively control the balance between these two objectives in the SF-CCA model.

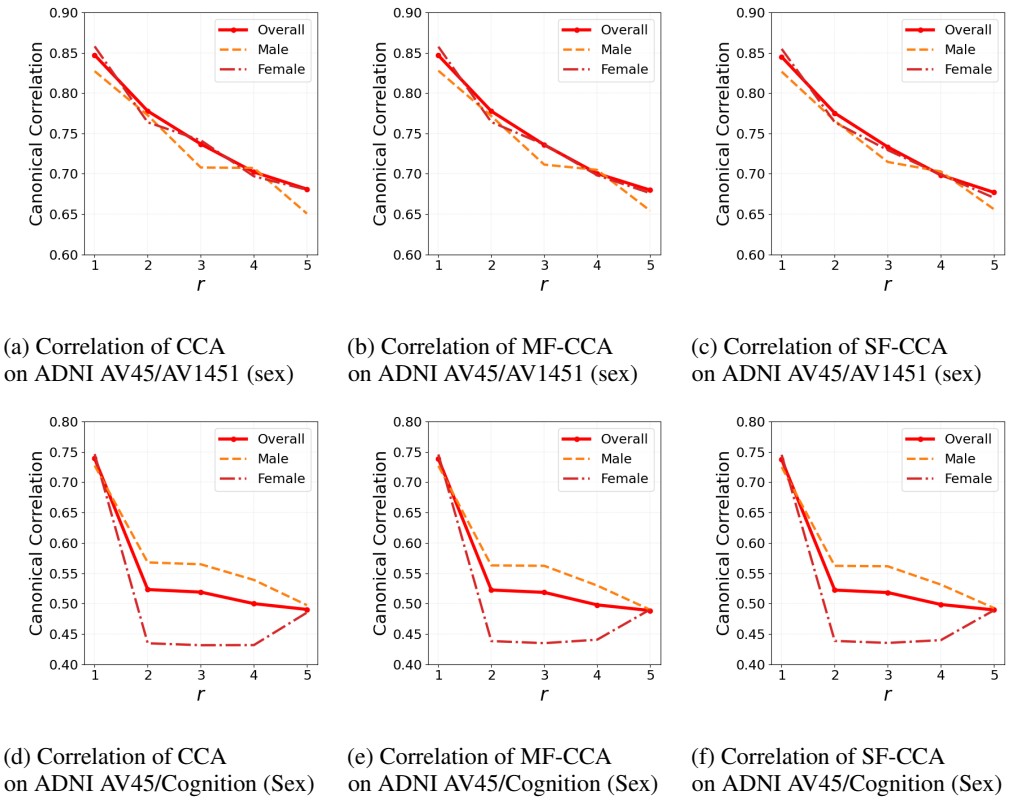

(a) Correlation of CCA
on ADNI AV45/AV1451 (sex)

(b) Correlation of MF-CCA
on ADNI AV45/AV1451 (sex)

(c) Correlation of SF-CCA
on ADNI AV45/AV1451 (sex)

(d) Correlation of CCA
on ADNI AV45/Cognition (Sex)

(e) Correlation of MF-CCA
on ADNI AV45/Cognition (Sex)

(f) Correlation of SF-CCA
on ADNI AV45/Cognition (Sex)

Figure 10: Visualization of the canonical correlation results of ADNI for the total five projection dimensions ($r$). All the methods are applied to both the entire dataset and individual subgroups. The closer each subgroup's curve is to the overall curve, the better.

## B.3 Runtime Sensitivity of MF-CCA and SF-CCA

When it comes to the issue of running time, it is important to consider the trade-off between computational efficiency and hyperparameter tuning effort. While SF-CCA may require more effort in tuning its hyperparameters, it still demonstrates a notable advantage in terms of time complexity compared to MF-CCA. To conduct a sensitivity analysis concerning the sample count ($N$), the feature count ($d$), and the subgroup count($K$), we conduct a series of experiments on synthetic datasets, ensuring the hyperparameters remained constant. These experiments are performed in three distinct settings, each replicated ten times, employing techniques including CCA, MF-CCA, and SF-CCA. The specifics of outcomes are depicted in Figures 12, 13, and 14.

### B.3.1 Runtime Sensitivity of MF-CCA and SF-CCA to $d$

In the first experimental setup, with the sample size held constant at $N = 2000$ and the number of groups held constant at $K = 5$, the dimensionality of features is varied across the set [50, 100, 150, 200, 250, 300, 350, 400]. According to Figure 12, it is discerned that as feature dimensionality increases, the runtime of MF-CCA exhibits a concomitant augmentation. Conversely, the runtime associated with SF-CCA remains comparatively stable throughout the varying feature sizes.

### B.3.2 Runtime Sensitivity of MF-CCA and SF-CCA to $N$

In the subsequent experimental configuration, the feature size is held constant at $d = 100$, while the sample size varies across the set [600, 800, 1000, 1200, 1400, 1600, 1800, 2000]. Notably, according to Figure 13, the increment in the sample size exerts a minimal impact on the runtime duration of SF-CCA. This observation can be rationalized by considering that an increase in the number of features would lead to a covariance matrix of greater dimensionality. Consequently, the

Table 4: Numerical results regarding three metrics including Correlation ($\rho_r$), Maximum Disparity ($\Delta_{\max,r}$), and Aggregate Disparity ($\Delta_{\text{sum},r}$). The best ones in each row are bold, and the second best one is underlined. The analysis focuses on the initial five projection dimensions for NHANES and ADNI, the initial seven projection dimensions for Synthetic Data, and the initial two projection dimensions for MHAAPS. "↑" means the larger the better and "↓" means the smaller the better.

| Dataset | Dim. (r) | $\rho_r$ ↑ | | | $\Delta_{\max,r}$ ↓ | | | $\Delta_{\text{sum},r}$ ↓ | | |
|---|---|---|---|---|---|---|---|---|---|---|
| | | CCA | MF-CCA | SF-CCA | CCA | MF-CCA | SF-CCA | CCA | MF-CCA | SF-CCA |
| Synthetic Data | 1 | **0.8003** | 0.7934 | 0.7757 | 0.3013 | 0.2386 | **0.1829** | 2.8647 | 2.2836 | **1.7258** |
| | 2 | **0.7533** | 0.7475 | 0.7309 | 0.3555 | 0.2866 | **0.2241** | 3.3802 | 2.8119 | **2.2722** |
| | 3 | **0.7032** | 0.6980 | 0.6853 | 0.3215 | 0.2591 | **0.2033** | 3.0974 | 2.5135 | **1.9705** |
| | 4 | **0.5872** | 0.5818 | 0.5677 | 0.3784 | 0.2791 | **0.197** | 3.6619 | 2.6939 | **1.8134** |
| | 5 | **0.4717** | 0.4681 | 0.4581 | 0.4385 | 0.3313 | **0.2424** | 4.1649 | 3.1628 | **2.2304** |
| | 6 | **0.4186** | 0.4161 | 0.4131 | 0.1163 | 0.0430 | **0.0000** | 1.161 | 0.4142 | **0.0000** |
| | 7 | **0.4012** | 0.3989 | 0.3928 | 0.2156 | 0.1336 | **0.0488** | 2.0224 | 1.1771 | **0.4117** |
| NHANES (Education) | 1 | **0.7360** | 0.7305 | 0.7330 | 0.0149 | 0.0113 | **0.0092** | 0.0596 | 0.0450 | **0.0366** |
| | 2 | **0.6392** | 0.6360 | 0.6334 | 0.0485 | 0.0359 | **0.0245** | 0.1941 | 0.1435 | **0.0980** |
| | 3 | **0.6195** | 0.6163 | 0.6147 | 0.0423 | 0.0322 | **0.0187** | 0.1691 | 0.1287 | **0.0747** |
| | 4 | **0.5027** | 0.4961 | 0.4990 | 0.0506 | 0.0342 | **0.0243** | 0.2022 | 0.1367 | **0.0974** |
| | 5 | **0.4416** | 0.4393 | 0.4392 | 0.1001 | **0.0818** | 0.0824 | 0.4003 | **0.3272** | 0.3297 |
| NHANES (Race) | 1 | **0.7376** | 0.7359 | 0.7365 | 0.0482 | 0.0479 | **0.0468** | 0.1927 | 0.1916 | **0.1874** |
| | 2 | **0.6400** | 0.6374 | 0.6383 | 0.0101 | 0.0097 | **0.0048** | 0.0402 | 0.0389 | **0.0191** |
| | 3 | **0.6221** | 0.6216 | 0.6195 | 0.0739 | 0.0708 | **0.0608** | 0.2955 | 0.2834 | **0.2432** |
| | 4 | **0.5001** | 0.4990 | 0.4980 | 0.0887 | 0.0774 | **0.0685** | 0.3549 | 0.3096 | **0.2742** |
| | 5 | **0.4459** | 0.4454 | 0.4442 | 0.1561 | 0.1451 | **0.1413** | 0.6244 | 0.5805 | **0.5653** |
| MHAAPS (Sex) | 1 | **0.4464** | 0.4451 | 0.4455 | 0.0093 | 0.0076 | **0.0044** | 0.0187 | 0.0152 | **0.0088** |
| | 2 | **0.1534** | 0.1529 | 0.1526 | 0.0061 | 0.0038 | **0.0019** | 0.0122 | 0.0075 | **0.0039** |
| ADNI AV45 and AV1451 (Sex) | 1 | **0.8472** | 0.8468 | 0.8450 | 0.0190 | 0.0180 | **0.0165** | 0.038 | 0.036 | **0.0329** |
| | 2 | **0.7778** | 0.7776 | 0.7753 | 0.0131 | 0.0119 | **0.0064** | 0.0263 | 0.0238 | **0.0127** |
| | 3 | **0.7369** | 0.7360 | 0.7332 | 0.0460 | 0.0371 | **0.0269** | 0.0919 | 0.0743 | **0.0538** |
| | 4 | **0.7022** | 0.7003 | 0.6985 | 0.0083 | 0.0046 | **0.0018** | 0.0167 | 0.0092 | **0.0037** |
| | 5 | **0.6810** | 0.6798 | 0.6770 | 0.0477 | 0.0399 | **0.0324** | 0.0954 | 0.0799 | **0.0648** |
| ADNI AV45 and Cognition (Sex) | 1 | **0.7391** | 0.7386 | 0.7373 | 0.0146 | 0.0149 | **0.0135** | 0.0293 | 0.0299 | **0.0270** |
| | 2 | **0.5232** | 0.5224 | 0.5223 | 0.1212 | 0.1127 | **0.1116** | 0.2424 | 0.2254 | **0.2233** |
| | 3 | **0.5189** | 0.5185 | 0.5182 | 0.1568 | 0.1508 | **0.1497** | 0.3135 | 0.3017 | **0.2994** |
| | 4 | **0.4999** | 0.4979 | 0.4985 | 0.1242 | 0.1061 | **0.1084** | 0.2485 | 0.2123 | **0.2168** |
| | 5 | **0.4904** | 0.4886 | 0.4896 | 0.0249 | **0.0124** | 0.0165 | 0.0498 | **0.0248** | 0.0329 |

corresponding numerical computations, such as the resolution of eigenvalues, become increasingly time-intensive.

### B.3.3 Runtime Sensitivity of MF-CCA and SF-CCA to $K$

Finally, we examined the sensitivity of both SF-CCA and MF-CCA to the number of subgroups. Synthetic data was generated with varying numbers of subgroups, allowing us to assess the corresponding running time. The results are presented in Figure 14. From the figure, it is evident that conventional CCA is not significantly sensitive to the number of groups. However, MF-CCA exhibits a higher level of sensitivity to the number of groups compared to SF-CCA.

## C  Experimental Details and Hyperparameter Selection Procedure

In this section, we delve into the particulars of our experiments, which encompass the choice of the penalty function and hyperparameters. We initiate our discussion with the penalty function $\phi$ outlined in Equation (8). This function takes on various forms, such as absolute, square, and exponential functions. For our experimental purposes, we concentrate specifically on the absolute function.

The distinctive characteristic of the absolute function lies in its robustness against minor discrepancies, as opposed to the square function, which tends to converge rapidly towards zero. It's worth noting

Table 5: Performance change from baseline CCA to F-CCA algorithms (MF-CCA and SF-CCA) in the format of percentage regarding three metrics: Correlation ($\rho_r$), Maximum Disparity ($\Delta_{\max,r}$), and Aggregate Disparity ($\Delta_{\text{sum},r}$). The changes are computed using the numerical results from Table 4: $P\rho_r = \frac{\rho_r \text{ of F-CCA} - \rho_r \text{ of CCA}}{\rho_r \text{ of CCA}} \times 100$, $P\Delta_{\max,r} = -\frac{\Delta_{\max,r} \text{ of F-CCA} - \Delta_{\max,r} \text{ of CCA}}{\Delta_{\max,r} \text{ of CCA}} \times 100$, $P\Delta_{\text{sum},r} = -\frac{\Delta_{\text{sum},r} \text{ of F-CCA} - \Delta_{\text{sum},r} \text{ of CCA}}{\Delta_{\text{sum},r} \text{ of CCA}} \times 100$. Here, F-CCA is replaced with either MF-CCA or SF-CCA to obtain the results in each column. The formulations are determined based on the properties of the metrics where $\rho_r$ is the larger the better while $\Delta_{\max,r}$ and $\Delta_{\text{sum},r}$ are the smaller the better.

| Dataset | Dim. (r) | $P\rho_r$ MF-CCA | $P\rho_r$ SF-CCA | $P\Delta_{\max,r}$ MF-CCA | $P\Delta_{\max,r}$ SF-CCA | $P\Delta_{\text{sum},r}$ MF-CCA | $P\Delta_{\text{sum},r}$ SF-CCA |
|---|---|---|---|---|---|---|---|
| Synthetic Data | 1 | -0.8688 | -3.0817 | 20.8021 | 39.2877 | 20.2834 | 39.7572 |
| | 2 | -0.7726 | -2.9710 | 19.3625 | 36.9495 | 16.8132 | 32.7792 |
| | 3 | -0.7366 | -2.5461 | 19.3916 | 36.7677 | 18.8495 | 36.3813 |
| | 4 | -0.9243 | -3.3148 | 26.245 | 47.9246 | 26.4333 | 50.4784 |
| | 5 | -0.7660 | -2.8896 | 24.4426 | 44.7114 | 24.0612 | 46.4484 |
| | 6 | -0.5865 | -1.3126 | 63.0023 | 99.9997 | 64.3202 | 99.9997 |
| | 7 | -0.5767 | -2.0838 | 38.0292 | 77.3821 | 41.7986 | 79.6436 |
| NHANES (Education) | 1 | -0.7511 | -0.4184 | 24.5248 | 38.5621 | 24.5248 | 38.5621 |
| | 2 | -0.5131 | -0.9070 | 26.0899 | 49.5206 | 26.0899 | 49.5206 |
| | 3 | -0.5243 | -0.7835 | 23.8580 | 55.7921 | 23.8580 | 55.7921 |
| | 4 | -1.3118 | -0.7309 | 32.4151 | 51.833 | 32.4151 | 51.833 |
| | 5 | -0.5175 | -0.5422 | 18.2615 | 17.6268 | 18.2615 | 17.6268 |
| NHANES (Race) | 1 | -0.2329 | -0.1450 | 0.5481 | 2.7539 | 0.5481 | 2.7539 |
| | 2 | -0.3972 | -0.2625 | 3.3229 | 52.6573 | 3.3229 | 52.6573 |
| | 3 | -0.0931 | -0.4209 | 4.1097 | 17.7082 | 4.1097 | 17.7082 |
| | 4 | -0.2334 | -0.4317 | 12.7708 | 22.7437 | 12.7708 | 22.7437 |
| | 5 | -0.1113 | -0.3914 | 7.0315 | 9.4561 | 7.0315 | 9.4561 |
| MHAAPS (Sex) | 1 | -0.2917 | -0.2084 | 18.6150 | 52.8984 | 18.6150 | 52.8984 |
| | 2 | -0.2724 | -0.4941 | 38.2692 | 68.1768 | 38.2692 | 68.1768 |
| ADNI AV45 and AV1451 (Sex) | 1 | -0.0444 | -0.2604 | 5.3213 | 13.4811 | 5.3213 | 13.4811 |
| | 2 | -0.0222 | -0.3178 | 9.5217 | 51.4446 | 9.5217 | 51.4446 |
| | 3 | -0.1294 | -0.4999 | 19.1779 | 41.4136 | 19.1779 | 41.4136 |
| | 4 | -0.2648 | -0.5307 | 44.6624 | 77.8152 | 44.6624 | 77.8152 |
| | 5 | -0.1737 | -0.5918 | 16.288 | 32.0918 | 16.288 | 32.0918 |
| ADNI AV45 and Cognition (Sex) | 1 | -0.0631 | -0.2416 | -2.0189 | 7.7392 | -2.0189 | -7.7392 |
| | 2 | -0.1436 | -0.1690 | 7.0116 | 7.8657 | 7.0116 | 7.8657 |
| | 3 | -0.0832 | -0.1386 | 3.7799 | 4.5125 | 3.7799 | 4.5125 |
| | 4 | -0.4074 | -0.2842 | 14.5679 | 12.7538 | 14.5679 | 12.7538 |
| | 5 | -0.3663 | -0.1533 | 50.094 | 33.8674 | 50.094 | 33.8674 |

that due to the non-differentiability of the absolute function at the origin, its identification within the MATLAB environment requires the utilization of the `sign` function. When the input to the penalty function reaches zero, the `sign` function indicates that the discrepancy error has already reached zero, leading to the termination of the training process.

### C.1 Hyperparameters Selection

When it comes to selecting hyperparameters, the key decisions revolve around providing the appropriate learning rate and the parameter $\lambda$ within the SF-CCA framework. Our approach consists of a two-step process. Firstly, we ascertain the optimal $\lambda$ for each dataset through an extensive sensitivity analysis, as detailed in Section B.2.3. With these derived values in hand, we then conduct a comprehensive grid search to determine the best-fitting learning rate. The unique parameter combinations utilized for each experimental set are elucidated as follows:

- **Synthetic Data**: $\lambda = 10$, learning rate of 2e-2 in SF-CCA, and 4e-1 in MF-CCA.
- **NHANES (Education)**: $\lambda = 5$, learning rate of 2e-3 in SF-CCA, and 5e-2 in MF-CCA.

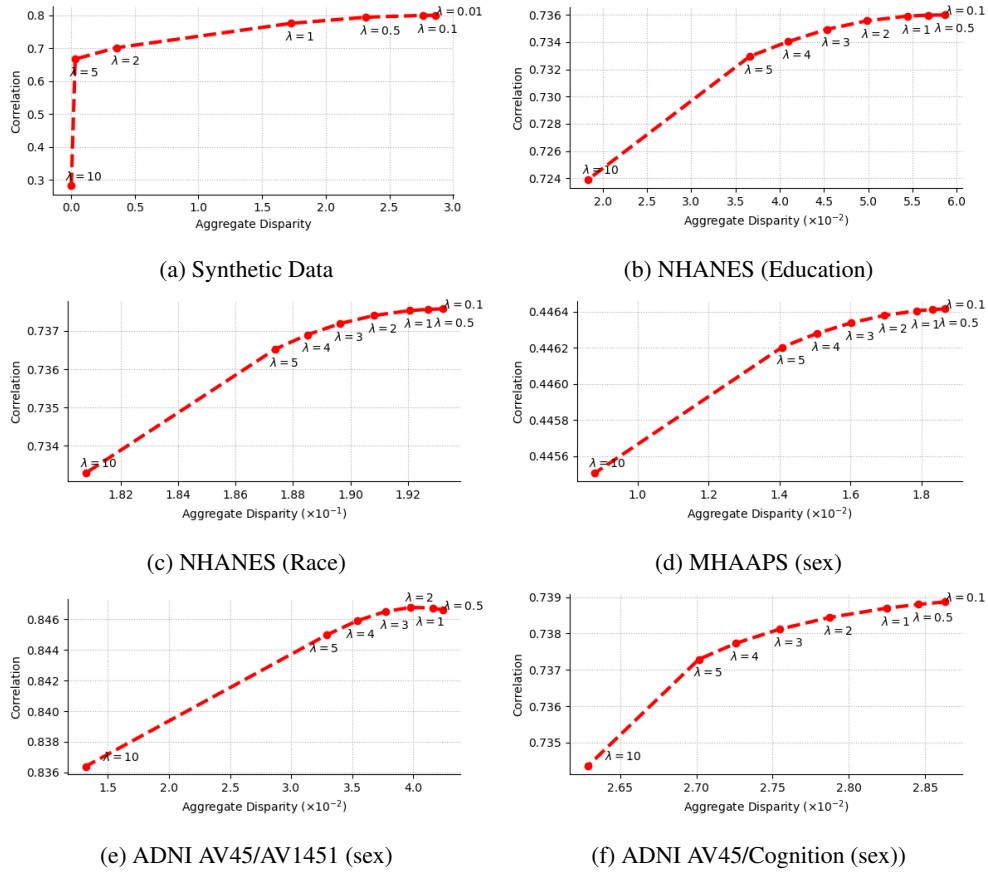

(a) Synthetic Data

(b) NHANES (Education)

(c) NHANES (Race)

(d) MHAAPS (sex)

(e) ADNI AV45/AV1451 (sex)

(f) ADNI AV45/Cognition (sex))

Figure 11: Sensitivity of correlation and disparity error to $\lambda$ in SF-CCA framework. Higher $\lambda$ emphasizes fairness over correlation (accuracy). Moving right to left, accuracy drops as fairness improves (smaller disparity). A notable trend links higher correlation with reduced fairness.

- **NHANES (Race)**: $\lambda = 5$, learning rate of 1e-3 in SF-CCA, and 2e-2 in MF-CCA.
- **MHAAPS**: $\lambda = 10$, learning rate of 2e-2 in SF-CCA, and 4e-1 in MF-CCA.
- **ADNI (AV45 and AV1451)**: $\lambda = 5$, learning rate of 1e-3 in SF-CCA, and 1e-2 in MF-CCA.
- **ADNI (AV45 and Cognition)**: $\lambda = 5$, learning rate of 1e-3 in SF-CCA, and 2e-2 in MF-CCA.

## C.2 Fairness and Correlation Measures

The correlation and fairness between $\mathbf{X}$ and $\mathbf{Y}$ under projections of $\mathbf{U}$ and $\mathbf{V}$ within $R$-dimensional spaces can be quantitatively measured by

$$\rho = \frac{\text{trace}(\mathbf{U}^\top \mathbf{X}^\top \mathbf{Y} \mathbf{V})}{\sqrt{\text{trace}(\mathbf{U}^\top \mathbf{X}^\top \mathbf{X} \mathbf{U}) \text{trace}(\mathbf{V}^\top \mathbf{Y}^\top \mathbf{Y} \mathbf{V})}}, \tag{21a}$$

$$\Delta_{\max} = \max_{i,j \in [K]} \left| \mathcal{E}^i(\mathbf{U}, \mathbf{V}) - \mathcal{E}^j(\mathbf{U}, \mathbf{V}) \right|, \tag{21b}$$

$$\Delta_{\text{sum}} = \sum_{i,j \in [K]} \left| \mathcal{E}^i(\mathbf{U}, \mathbf{V}) - \mathcal{E}^j(\mathbf{U}, \mathbf{V}) \right|. \tag{21c}$$

In Section 4.1, instead of employing matrix-based measurements (21), we adopted component-wise measurements (13) to facilitate detailed observations on each individual projection dimension $r$. In the following discussion, we demonstrate that the component-wise measurements, across all $r$ dimensions, provide a more aggressive measurement approach compared to the matrix variants (21). In other words, it is evident that having large component-wise correlation values $\rho_r$ and small

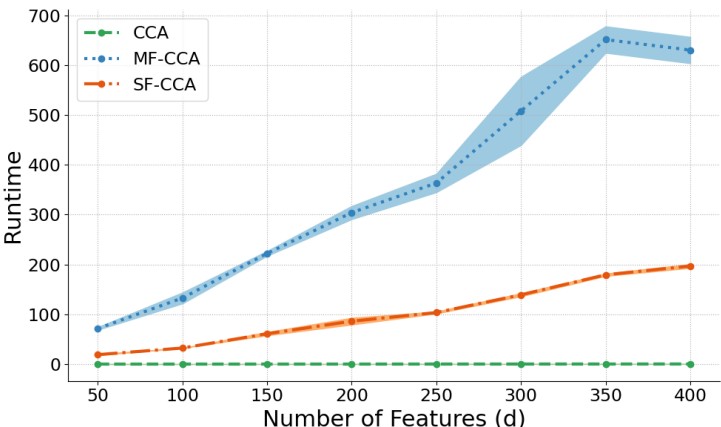

Figure 12: Computation time (mean±std) of 10 repeated experiments for the total three projection dimensions on synthetic data comprising four subgroups ($K$). The number of samples is fixed at $N = 2000$, while the number of features varies.

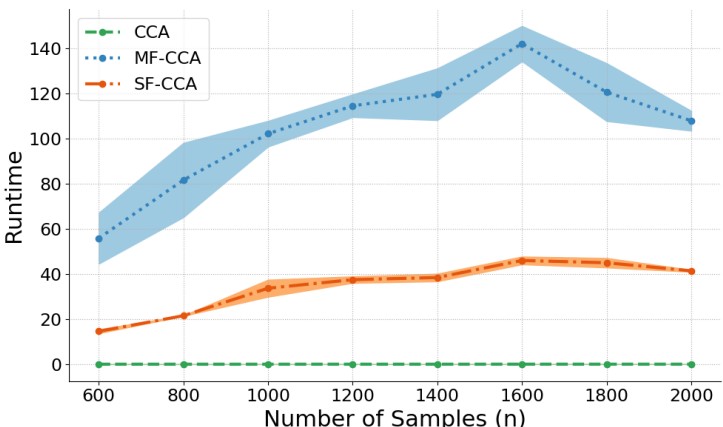

Figure 13: Computation time (mean±std) of 10 repeated experiments for the total three projection dimensions on synthetic data comprising four subgroups ($K$). The number of features is fixed at $d = 100$, and the number of groups is held constant at $K = 5$, while the number of samples varies.

disparity errors $\Delta_{\mathrm{max},r}$ and $\Delta_{\mathrm{sum},r}$ for all $r \in [R]$ can guarantee large values of $\rho$ and small values of $\Delta_{\mathrm{max}}$ and $\Delta_{\mathrm{sum}}$.

The following lemma rigorously supports this observation.

**Lemma 13.** *Let* $\rho_r, \Delta_{\mathrm{max},r}, \Delta_{\mathrm{sum},r}$ *be defined as in* (13), *and let* $\rho, \Delta_{\mathrm{max}}, \Delta_{\mathrm{sum}}$ *be defined as in* (21). *Then,*

$$\rho = \frac{1}{R}\sum_{r=1}^{R}\rho_r, \quad \Delta_{\mathrm{max}} \le \sum_{r=1}^{R}\Delta_{\mathrm{max},r}, \quad \Delta_{\mathrm{sum}} \le \sum_{r=1}^{R}\Delta_{\mathrm{sum},r}.$$

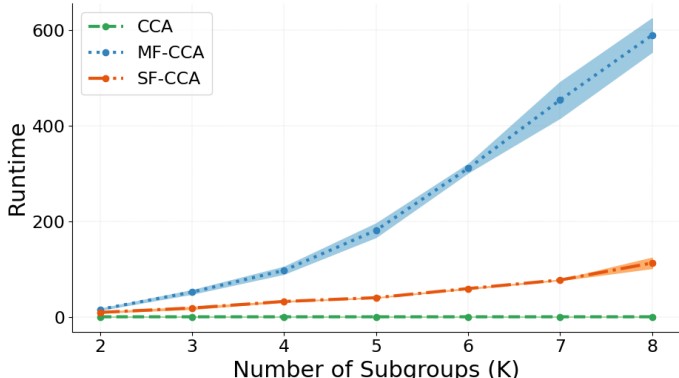

Figure 14: Computation time (mean±std) of 10 repeated experiments for the total seven projection dimensions on synthetic data comprising varying numbers of subgroups ($K$). The number of features is fixed at $d = 100$.

*Proof.* Since $\mathbf{U} = [\mathbf{u}_1, \cdots, \mathbf{u}_R] \in \mathbb{R}^{D_x \times R}$ and $\mathbf{V} = [\mathbf{v}_1, \cdots, \mathbf{v}_R] \in \mathbb{R}^{D_y \times R}$, we have

$$
\begin{aligned}
\text{trace}(\mathbf{U}^\top \mathbf{X}^\top \mathbf{Y} \mathbf{V}) &= \sum_{r=1}^{R} \mathbf{u}_r^\top \mathbf{X}^\top \mathbf{Y} \mathbf{v}_r, \\
\text{trace}(\mathbf{U}^\top \mathbf{X}^\top \mathbf{X} \mathbf{U}) &= \sum_{r=1}^{R} \mathbf{u}_r^\top \mathbf{X}^\top \mathbf{X} \mathbf{u}_r, \\
\text{trace}(\mathbf{V}^\top \mathbf{Y}^\top \mathbf{Y} \mathbf{V}) &= \sum_{r=1}^{R} \mathbf{v}_r^\top \mathbf{Y}^\top \mathbf{Y} \mathbf{v}_r.
\end{aligned}
\tag{22}
$$

Note that $\mathcal{U} = \{\mathbf{U} | \mathbf{U}^\top \mathbf{X}^\top \mathbf{X} \mathbf{U} = \mathbf{I}_R\}$ and $\mathcal{V} = \{\mathbf{V} | \mathbf{V}^\top \mathbf{Y}^\top \mathbf{Y} \mathbf{V} = \mathbf{I}_R\}$. Using the implementation of the retraction operation on $\mathbf{U}$ and $\mathbf{V}$ (see, (19) and (17)), we obtain

$$
\begin{aligned}
\mathbf{u}_r \mathbf{X}^\top \mathbf{X} \mathbf{u}_r &= \mathbf{v}_r \mathbf{Y}^\top \mathbf{Y} \mathbf{v}_r = 1, \\
\mathbf{U}^\top \mathbf{X}^\top \mathbf{X} \mathbf{U} &= \mathbf{V}^\top \mathbf{Y}^\top \mathbf{Y} \mathbf{V} = \mathbf{I}_R, \\
\text{trace}(\mathbf{U}^\top \mathbf{X}^\top \mathbf{X} \mathbf{U}) \text{trace}(\mathbf{V}^\top \mathbf{Y}^\top \mathbf{Y} \mathbf{V}) &= \text{trace}(\mathbf{I}_R)^2 = R^2.
\end{aligned}
\tag{23}
$$

Thus, for the component-wise measure $\rho_r$ defined in (13a) and the matrix variant $\rho$ defined in (21a), we have

$$
\begin{aligned}
\rho &= \frac{\text{trace}(\mathbf{U}^\top \mathbf{X}^\top \mathbf{Y} \mathbf{V})}{\sqrt{\text{trace}(\mathbf{U}^\top \mathbf{X}^\top \mathbf{X} \mathbf{U}) \text{trace}(\mathbf{V}^\top \mathbf{Y}^\top \mathbf{Y} \mathbf{V})}} \\
&= \frac{\sum_{r=1}^{R} \mathbf{u}_r^\top \mathbf{X}^\top \mathbf{Y} \mathbf{v}_r}{\sqrt{R^2}} \\
&= \frac{1}{R} \sum_{r=1}^{R} \frac{\mathbf{u}_r^\top \mathbf{X}^\top \mathbf{Y} \mathbf{v}_r}{1} \\
&= \frac{1}{R} \sum_{r=1}^{R} \frac{\mathbf{u}_r^\top \mathbf{X}^\top \mathbf{Y} \mathbf{v}_r}{\sqrt{\mathbf{u}_r^\top \mathbf{X}^\top \mathbf{X} \mathbf{u}_r \mathbf{v}_r^\top \mathbf{Y}^\top \mathbf{Y} \mathbf{v}_r}} \\
&= \frac{1}{R} \sum_{r=1}^{R} \rho_r.
\end{aligned}
\tag{24}
$$

For component-wise measure $\Delta_{\max,r}$ and matrix measure $\Delta_{\max}$, we have

$$
\begin{aligned}
\Delta_{\max} &= \max_{i,j\in[K]} \left| \mathcal{E}^i(\mathbf{U},\mathbf{V}) - \mathcal{E}^j(\mathbf{U},\mathbf{V}) \right| \\
&= \max_{i,j\in[K]} \left| \left( \operatorname{trace}\left( \mathbf{U}^{i,\star^\top}\mathbf{X}^{i^\top}\mathbf{Y}^i\mathbf{V}^{i,\star} \right) - \operatorname{trace}\left( \mathbf{U}^\top\mathbf{X}^{i^\top}\mathbf{Y}^i\mathbf{V} \right) \right) \right. \\
&\qquad\qquad \left. - \left( \operatorname{trace}\left( \mathbf{U}^{j,\star^\top}\mathbf{X}^{j^\top}\mathbf{Y}^j\mathbf{V}^{j,\star} \right) - \operatorname{trace}\left( \mathbf{U}^\top\mathbf{X}^{j^\top}\mathbf{Y}^j\mathbf{V} \right) \right) \right| \\
&= \max_{i,j\in[K]} \left| \left( \sum_{r=1}^R \mathbf{u}_r^{i,*^\top}\mathbf{X}^{i^\top}\mathbf{Y}^i\mathbf{v}_r^{i,*} - \sum_{r=1}^R \mathbf{u}_r^\top\mathbf{X}^{i^\top}\mathbf{Y}^i\mathbf{v}_r \right) \right. \\
&\qquad\qquad \left. - \left( \sum_{r=1}^R \mathbf{u}_r^{j,*^\top}\mathbf{X}^{j^\top}\mathbf{Y}^j\mathbf{v}_r^{j,*} - \sum_{r=1}^R \mathbf{u}_r^\top\mathbf{X}^{j^\top}\mathbf{Y}^j\mathbf{v}_r \right) \right| \\
&= \max_{i,j\in[K]} \left| \sum_{r=1}^R \left( \left( \mathbf{u}_r^{i,*^\top}\mathbf{X}^{i^\top}\mathbf{Y}^i\mathbf{v}_r^{i,*} - \mathbf{u}_r^\top\mathbf{X}^{i^\top}\mathbf{Y}^i\mathbf{v}_r \right) \right. \right. \\
&\qquad\qquad \left. \left. - \left( \mathbf{u}_r^{j,*^\top}\mathbf{X}^{j^\top}\mathbf{Y}^j\mathbf{v}_r^{j,*} - \mathbf{u}_r^\top\mathbf{X}^{j^\top}\mathbf{Y}^j\mathbf{v}_r \right) \right) \right| \\
&\le \sum_{r=1}^R \max_{i,j\in[K]} \left| \left( \mathbf{u}_r^{i,*^\top}\mathbf{X}^{i^\top}\mathbf{Y}^i\mathbf{v}_r^{i,*} - \mathbf{u}_r^\top\mathbf{X}^{i^\top}\mathbf{Y}^i\mathbf{v}_r \right) \right. \\
&\qquad\qquad \left. - \left( \mathbf{u}_r^{j,*^\top}\mathbf{X}^{j^\top}\mathbf{Y}^j\mathbf{v}_r^{j,*} - \mathbf{u}_r^\top\mathbf{X}^{j^\top}\mathbf{Y}^j\mathbf{v}_r \right) \right| \\
&= \sum_{r=1}^R \max_{i,j\in[K]} \left| \mathcal{E}^i(\mathbf{u}_r,\mathbf{v}_r) - \mathcal{E}^j(\mathbf{u}_r,\mathbf{v}_r) \right| \\
&= \sum_{r=1}^R \Delta_{\max,r}.
\end{aligned}
\tag{25}
$$

Similarly, for $\Delta_{\mathrm{sum},r}$ and $\Delta_{\mathrm{sum}}$, we have $\Delta_{\mathrm{sum}} \le \sum_{r=1}^R \Delta_{\mathrm{sum},r}$. $\qquad\square$

