# OpenReview forum: "Fair Canonical Correlation Analysis"
_NeurIPS.cc/2023/Conference — NeurIPS 2023 poster_

### Official Review · Reviewer_6yqF · 2023-06-28

**Soundness:** 3 good
**Presentation:** 3 good
**Contribution:** 3 good
**Rating:** 6
**Confidence:** 4

**Summary:**

This paper proposes a fair CCA algorithm that aims to find fair CCA projection matrices. The authors claim that it is necessary to develop an appropriate algorithm for CCA with fairness guarantee. In the presence of sensitive attribute and unfairness in observed data, the proposed algorithms successfully reduces certain unfairness of the learned projection matrices. Specifically, the two proposed algorithms, MF-CCA and SF-CCA, are optimized using gradient descent algorithms on Stiefel manifolds. The convergences of these algorithms are also theoretically guaranteed, and several experiments conducted provide empirical support for the proposed algorithms.

**Strengths:**

- The paper is generally well-written and easy to follow in overall.
- To the best of my knowledge, this work is the first to address the issue of unfairness issue in CCA, which could be a milestone.
- The fairness metric targeted (i.e., Correlation Disparity Error in Definition 1) is well-defined and fits well with other general fairness notions used in fair prediction tasks (e.g., demographic parity).
- The proposed algorithms (MF-CCA and SF-CCA) align well with the theoretical studies presented (Theorems 4 and 5).
- The roles of the two algorithms well-specified. MF-CCA finds the optimal solution by minimizing CCA error and unfairness losses simultaneously, while SF-CCA provides an advantage in controlling the trade-off between CCA error and unfairness losses.
- Empirical results validate the efficacy of MF-CCA and SF-CCA, not only on synthetic data but also on real datasets.

**Weaknesses:**

- There is no theoretical guarantee for fairness provided. If the authors could theoretically demonstrate that the solutions to equations 7 and 9 have low $\mathcal{E}^{k} (U, V),$ as has already been empirically shown, the contribution would be more novel.
- To illustrate the ability of SF-CCA in controlling the trade-off between error and fairness, a visualization such as Pareto-front lines (commonly used in fair classification problems [1, 2]) would be beneficial. Table 1 only presents results using a single $\lambda$ selected from the set [1e-2, 1e-1, 1, 10, 100].

[1] https://arxiv.org/abs/1802.06309

[2] https://arxiv.org/abs/2103.06503


**Questions:**

- Which penalty function, $\phi$, was used in the experiments?
- Is the computation time more significantly impacted by the size of the training data or the dimension of the input feature?
- Could the authors provide the definition of a 'componentwise Lipschitz continuous function' as stated in Assumption A?

**Limitations:**

- Unlike the vanilla CCA which solves the objective by optimizing two matrices, $U$ and $V$, the proposed algorithms require more matrices to be optimized, the number of which increases with the number of sensitive attributes, $K.$
- Naturally, the computational cost of these proposed algorithms is higher than that of the vanilla CCA due to the fairness objectives they must minimize. Any future work aimed at reducing this computational cost would certainly be novel.

---

> ### Author Rebuttal · Authors · 2023-08-09
>
> > **W1:** There is no theoretical guarantee for fairness provided. If the authors could theoretically demonstrate that the solutions to equations 7 and 9 have low \(\epsilon^k(\mathbf{U},\mathbf{V})\) as has already been empirically shown, the contribution would be more novel.
>
> **Response:** Thank you for your comment.  The CCA problem's similarity to PCA in its non-convex nature means that Theorem 4 (multi-objective) causes the Pareto descent direction norm to diminish, converging algorithm solutions to stationary fair subspaces. To make the CCA objective convex, we can introduce regularization such as $\ell_2$-sqaure type penalty with hyperparameter $\alpha>0$,  denoted as $R_{\alpha}(\mathbf{U},\mathbf{V})$, that maintains positive semi-definite Hessian matrices of objectives. This adjustment aims to minimize the fairness error $\tilde{f}(\mathbf{U},\mathbf{V}) - \tilde{f}(\mathbf{U}^{\star},\mathbf{V}^{\star})$ toward zero, where $\tilde{f}(\mathbf{U},\mathbf{V}) = f(\mathbf{U},\mathbf{V}) + R_{\alpha}(\mathbf{U},\mathbf{V})$, and $f$ represents multi or single objectives as in (7) or (9) respectively. However, the regularization-based approach requires more hyperparameters, leading us to primarily focus on the standard non-convex method in CCA literature. Thus, we offer theoretical first-order stationary analysis using $\left\Vert \mathbf{P} \right\Vert$ and experimental evidence of algorithmic convergence toward fair subspaces.
>
>
> > **W2:**  To illustrate the ability of SF-CCA in controlling the trade-off between error and fairness, a visualization such as Pareto-front lines (commonly used in fair classification problems [1, 2]) would be beneficial. Table 1 only presents results using a single selected from the set $ \lambda$ in $ [1e-2, 1e-1, 1, 10, 100]$.
>
> **Response:** Thank you for your valuable comment.
>
> -  Our experiments examined the influence of the $\lambda$ parameter in SF-CCA on correlation and disparity (Figure 12 in the appendix). As expected, higher $\lambda$ values led to decreased disparity and correlation. Notably, a gradual decline in correlation was observed with increasing $\lambda$, while disparity decreased rapidly. This underscores our framework's ability to boost fairness without significant accuracy loss. In Figure 12 (a), correlation plateaued between $\lambda$ values of 0.01 and 10, while disparity swiftly neared zero at $\lambda = 1$. Even at $\lambda = 10$, the correlation remained reasonable. Importantly, the optimal correlation-fairness balance was attained at $\lambda = 1$.
>
> -  We extended our experiments to real datasets, illustrated in Figure 2 of the attached PDF. A consistent pattern emerged: as fairness improved (disparity decreased), accuracy (correlation) declined. Yet, by pinpointing an optimal $\lambda$, we significantly improved fairness without compromising accuracy.
>
> Hence, our approach attains fairness with competitive accuracy, distinguishing it from traditional CCA methods that overlook fairness concerns.
>
>
> > **Q1:**  Is the computation time more significantly impacted by the size of the training data or the dimension of the input feature?
>
> **Response:** We've presented time computations for algorithms on real and synthetic datasets in Table 2. CCA's efficiency stems from its singular eigenvalue decomposition, while MF-CCA requires extensive gradient direction searches and repeats of gradient descent. SF-CCA, with a fixed trade-off parameter, runs faster than MF-CCA despite hyperparameter tuning. While comparing MF-CCA and SF-CCA is challenging due to their trade-offs, the extra time invested aligns with our goal of balancing fairness enhancement with accuracy retention.
>
> Further examining time complexity, we assessed SF-CCA and MF-CCA sensitivity to subgroup count. Figure 11 in Appendix B.3 reveals MF-CCA's increased sensitivity to subgroup count compared to SF-CCA. This corresponds to their performance in Table 2, where SF-CCA excelled in computational efficiency. SF-CCA's trade-offs are apparent: it achieves a balance between computational efficiency and fairness, while MF-CCA guarantees stronger fairness at the cost of longer computation times for more subgroups.
>
> ​​For sensitivity analysis on sample count ($n$) and feature count ($d$), we conducted two sets of  news experiments on synthetic data with fixed hyperparameters. Each experiment, repeated 20 times using CCA, MF-CCA, and SF-CCA, is detailed in Figure 1 of the attached PDF.
>
> - For the first set, we maintained sample size and varied feature size [50, 100, 150, 200, 250, 300, 350, 400]. MF-CCA exhibited extended runtimes with increased features, while SF-CCA's runtime remained relatively steady.
>
> - In the second set, fixing feature size, we varied sample size [600, 800, 1000, 1200, 1400, 1600, 1800, 2000]. The impact of   larger sample size on SF-CCA's runtime was relatively minimal.
>
>
> > **Q2:** Which penalty function, $\phi$ was used in the experiments?
>
> **Response:** Thank you for pointing this out. The function $\phi$ can take various forms, including absolute, square, exponential, and more. In our experiments, we specifically focus on the absolute function. Its strength lies in its resilience to minor disparities, unlike the square function, which tends to rapidly approach zero. We will ensure that these specific details are included in the paper for improved clarity.
>
> > **Q3:** Could the authors provide the definition of a 'componentwise Lipschitz continuous function' as stated in Assumption A?
>
> **Response:** Thank you for your question. A componentwise Lipschitz continuity means that each $\nabla f_i$ for $i \in [M]$ has Lipschitz continuity on the manifold $\mathcal{M}$ with constant $L_{i,M}$, and $\nabla \mathbf{F}$ is componentwise Lipschitz continuous on $\mathcal{M}$ with constant $L_F:=\max_{i=1,\ldots, M} L_{i,M}$. We will clarify this in Assumption A.

---

> > ### Comment · Reviewer_6yqF · 2023-08-17
> >
> > Thank you for your detailed responses.
> > Most of my concerns/questions have been addressed.
> >
> > - (W1) - Addressed.
> > Thank you for the clarification.
> >
> > - (W2) - Addressed.
> > I appreciate your efforts in providing the results of additional experiments.
> > It might be beneficial to include Figure 2 in the PDF (and even Figure 3 in the PDF), possibly along with Figure 12.
> > Considering this work as a milestone, these trade-off comparisons could serve as baselines if a new study related to fair CCA appears.
> >
> > - (Q1)
> > Thank you for providing additional experiments regarding sensitivity analysis on $n$ and $d.$
> > I think this point could be interpreted as a limitation of MF-CCA, representing a trade-off between achieving almost perfect fairness and computational time.
> >    - (New question) I believe that not only MF-CCA but also SF-CCA with a sufficiently large $\lambda$ could achieve (almost perfect) fairness.
> > However, the computation time of SF-CCA is lower than that of MF-CCA.
> > Given this context, what advantages does MF-CCA offer compared to SF-CCA?
> >
> > - (Q2, Q3) - Addressed.

---

> > > ### Author Response · Authors · 2023-08-18
> > > **Response to Reviewer 6yqF: MF-CCA vs. SF-CCA**
> > >
> > > > **New Question:** I believe that not only MF-CCA but also SF-CCA with a sufficiently large $\lambda$ could achieve (almost perfect) fairness. However, the computation time of SF-CCA is lower than that of MF-CCA. Given this context, what advantages does MF-CCA offer compared to SF-CCA?
> > >
> > > **Response:** Thank you for your great question.  SF-CCA simplifies optimization, reduces computational demand, and controls fairness-accuracy trade-offs through $\lambda$ adjustments. MF-CCA, on the other hand, offers noteworthy advantages, as detailed below:
> > >
> > > **I.** *Hyperparameter Search-Free:*  MF-CCA operates without hyperparameters, automatically identifying a Pareto stationary point.  SF-CCA, on the other hand,  requires tuning $\lambda$, which can be complex and contingent on dataset and application specifics. As illustrated in Figure 2 (attached to the [general response](https://openreview.net/forum?id=W3cDd5xlKZ&noteId=OZk945tKNu)):
> > >
> > >    - Synthetic data achieves fairness stability around $\lambda=2$.
> > >    - MHAAPS data reaches correlation stability at $\lambda=10^{-2}$.
> > >    - NHANES data demonstrates correlation stability near $\lambda=10^{-1}$.
> > >
> > > These instances emphasize the varying $\lambda$ search range across datasets, demanding substantial fine-tuning efforts.
> > >
> > > Expanding the $\lambda$ interval may seem like a solution, yet a **larger $\lambda$ could drastically reshape** the optimization landscape, possibly necessitating increased iterations to minimize the modified objective. For example, in our experiment, with $\lambda=100$, SF-CCA yielded a disparity error of $\geq 0.1$, whereas it was $\leq 0.0001$ with $\lambda=10$ using the same number of iterations.
> > >
> > > **II.** *Robustness and Flexibility:* MF-CCA allows us to adjust the relative weights assigned to different fairness objectives (i.e., $f_2 \ldots f_M$). This is important for dealing with imbalanced data, where different groups may have different levels of representation. In contrast, SF-CCA depends solely on $\lambda$ and can be suboptimal in imbalanced data.
> > >
> > > **III.** *Adaptive Fairness Trade-offs:* Achieving perfect fairness in every situation might not always be feasible or desirable. MF-CCA finds a Pareto stationary point that strikes an appropriate balance between fairness and accuracy. This adaptability is crucial when overly strict fairness constraints could lead to suboptimal performance in other critical aspects of the model.
> > >
> > > **IV.** *Balancing Diverse Fairness Metrics:* In reality, fairness can span multiple dimensions, including metrics like demographic parity, equalized odds, and group sufficiency. MF-CCA can address these objectives together, achieving well-rounded fairness across dimensions. Especially useful when SF-CCA with a single regularization parameter can't reconcile complex fairness concepts.
> > >
> > >
> > >  **Table 1:** Comparison of MF-CCA and SF-CCA
> > >
> > > | Feature | MF-CCA | SF-CCA |
> > > |---|---|---|
> > > | Hyperparameters | No | Yes (λ) |
> > > | Fairness-accuracy trade-off | Automatic | Controlled by λ |
> > > | Flexibility | Can adjust weights for different fairness objectives | Depends solely on λ |
> > > | Adaptability | Finds a Pareto stationary point | Can be suboptimal in imbalanced data |
> > > | Ability to balance diverse fairness metrics | Yes | Requires additional hyperparameters |
> > >
> > >
> > >
> > > Table 1 summarizes the comparison between MF-CCA and SF-CCA. We will include this comparison in the final revision.

---

### Official Review · Reviewer_1Xpv · 2023-07-06

**Soundness:** 4 excellent
**Presentation:** 2 fair
**Contribution:** 3 good
**Rating:** 7
**Confidence:** 2

**Summary:**

This paper addresses fairness and bias in Canonical Correlation Analysis (CCA). The authors propose a framework that minimizes correlation disparities associated with protected attributes, reducing unfairness without compromising accuracy. Experimental evaluation validates the effectiveness of the approach. The findings emphasize the importance of fairness in CCA applications.

**Strengths:**

- Novel Contribution: The paper introduces a framework to address fairness and bias concerns in Canonical Correlation Analysis (CCA), making a valuable contribution to the field.

- Practical Relevance: By focusing on CCA, a widely used statistical technique, the paper addresses a real-world problem and emphasizes the importance of considering fairness in data analysis.

- Experimental Validation: The authors conduct experiments on both synthetic and real-world datasets, providing empirical evidence of the effectiveness of their proposed framework in reducing unfairness without compromising the accuracy of the CCA model.

- Clear Presentation: The abstract provides a concise overview of the paper's objectives, approach, and findings, making it easy to understand the key contributions of the research.

**Weaknesses:**

- Lack of Detailed Metrics: The paper could benefit from providing more specific details about the metrics used to measure unfairness and bias in CCA. This would enhance the transparency and reproducibility of the experimental evaluation.

- Limited Comparison: The paper does not explicitly compare the proposed framework with existing fairness-aware CCA methods. Including such comparisons would provide insights into the relative performance and advantages of the proposed approach.

- Scope and Generalizability: While the paper addresses fairness concerns in CCA, the focus is limited to this specific technique. It would be beneficial to discuss the potential implications of the findings for other statistical methods or broader machine learning applications.

**Questions:**

- In the experimental setup, what considerations were made in selecting the synthetic and real-world datasets? Were there any specific characteristics of these datasets that influenced the results or generalizability of the findings?

- While the paper focuses on fairness and bias in CCA, could you elaborate on the potential implications of the findings for other statistical methods or broader machine learning applications? How transferable do you believe the proposed framework is beyond the scope of CCA?

- Are there any additional factors or considerations that should be taken into account when applying the proposed framework in practical settings? For instance, how would the framework handle missing data, outliers, or high-dimensional datasets?

- Given the goal of reducing unfairness, how does the proposed framework balance the trade-off between fairness and overall predictive accuracy? Were there any cases in the experiments where the framework significantly sacrificed accuracy to achieve fairness?

**Limitations:**

- Fairness Metrics: The paper could provide a more detailed discussion on the fairness metrics used to evaluate the proposed framework. Further elaboration on the choice and justification of these metrics would enhance the clarity and interpretability of the experimental results.

- Generalizability: The generalizability of the findings may be limited by the specific characteristics and distribution of the datasets used in the experiments. The authors could discuss the potential challenges or variations that may arise when applying the framework to other datasets or domains.

- Trade-off between Fairness and Accuracy: The paper briefly mentions that the proposed framework minimizes unfairness without compromising the accuracy of the CCA model. However, a more in-depth analysis of the potential trade-off between fairness and accuracy would provide a clearer understanding of the framework's limitations and its impact on prediction performance.

- Real-world Application Challenges: While the framework demonstrates effectiveness in reducing unfairness, the paper does not extensively discuss the practical challenges that may arise when applying the proposed approach to real-world scenarios, such as handling missing data, complex feature distributions, or scalability issues.

---

> ### Author Rebuttal · Authors · 2023-08-09
>
>
> > **Q1:**  In the experimental setup, what considerations … real-world datasets? Were there any specific characteristics of these datasets that influenced the results or generalizability of the findings?
>
> **Response:**  We carefully selected synthetic and real-world datasets to evaluate the fair CCA method across various scenarios. Real-world datasets from diverse fields further validated its applicability. More details about these datasets can be found in Appendix B. For example, ADNI data is analyzed for fairness in medical imaging classification [[ZYH22](https://arxiv.org/abs/2210.01725)]. Fairness concerns with ADNI image data could stem from under-representation of specific populations (ethnic, cultural, or economic backgrounds). Moreover, over- or under-representation of age, sex, or biometric groups might affect data fairness. Disease distribution among sensitivity features, such as higher AD occurrence in women, can also impact fairness [[MS16](https://www.thelancet.com/journals/laneur/article/PIIS1474-4422(16)00067-3/fulltext)].
>
> > **Q2:** While the paper focuses on fairness and bias in CCA, could … machine learning applications? How transferable do you believe the proposed framework is beyond the scope of CCA?
>
> **Response:**  Thank you for this valuable question.
> -  As suggested by reviewer MdZ1, our work holds potential for extension to multiple modalities. CCA has been adapted for intricate scenarios like multiset CCA, kernel CCA for nonlinear situations, and deep CCA for imaging data  [[ZYC20](https://pubmed.ncbi.nlm.nih.gov/32592530/)].
>
> - Our algorithm for smooth manifold multi-objective optimization can also tackle other problems like fair PCA, as both can be framed as optimization challenges on smooth manifolds [[Nicolas23](https://www.nicolasboumal.net/book/IntroOptimManifolds_Boumal_2023.pdf)].
>
>  -  As CCA finds applications in downstream tasks like clustering, classification, and manifold learning, our approach can effectively ensure fairness in such scenarios when utilizing CCA methods [[ZYC20](https://pubmed.ncbi.nlm.nih.gov/32592530/)].
>
> Thus, our framework has potential beyond CCA, but these extensions could introduce novel optimization and computation challenges, left for future research.
>
> > **Q3:** Are there any additional factors or considerations … in practical settings? For instance, how would the framework handle missing data, outliers, or high-dimensional datasets?
>
> **Response:**   Thanks for your question.
>
> - Extending fair CCA to handle missing data is notably more challenging than addressing missing data within standard CCA, given the potential for biases and distorted correlations. This challenge is emphasized in [[F21](https://onlinelibrary.wiley.com/doi/full/10.1002/int.22415),[ZL21](https://proceedings.neurips.cc/paper_files/paper/2021/hash/85dca1d270f7f9aef00c9d372f114482-Abstract.html)]. The added complexity arises from the integration of fairness considerations with missing data biases.
> - Outliers have the capacity to disrupt CCA, affecting reliability and interpretations, and potentially steering fair CCA toward suboptimal outcomes.
> - Sparse CCA addresses high-dimensional data using techniques like $L_1$ penalty to induce sparsity in canonical vectors. However, incorporating Sparse CCA into multi-objective optimization introduces challenges tied to nonsmooth optimization issues.
>
> In addressing these concerns, especially in our multi-objective context, we need novel formulations and analysis for future work.
> > **Q4:**  Given the goal of reducing unfairness, how does the proposed framework balance the trade-off between fairness and overall predictive accuracy?
>
> **Response:** Thank you for your question.
>
> -  Our experiments explored the impact of the $\lambda$ parameter in the SF-CCA method on correlation and disparity, as depicted in Figure 12 of the appendix. The results confirmed our expectations: higher $\lambda$ values led to reduced disparity and correlation. Interestingly, we observed a gradual decline in correlation as $\lambda$ increased, contrasted by a rapid reduction in disparity. This showcases our framework's ability to enhance fairness while preserving accuracy. In Figure 12 (a), the correlation plateaued between $\lambda$ values of 0.001 and 10, while disparity rapidly decreased to near-zero at $\lambda = 1$. Even at $\lambda = 10$, the correlation remained reasonable. Notably, the optimal balance between correlation and fairness was achieved at $\lambda = 1$.
>
> -  We extended our experiments to real datasets, illustrated in Figure 2 of the attached PDF in the general rebuttal. A consistent pattern emerged: as fairness improved (disparity decreased), accuracy (correlation) declined. Yet, by pinpointing an optimal $\lambda$, we significantly improved fairness without compromising accuracy.
>
> In summary, our approach attains fairness with competitive accuracy, distinguishing it from traditional CCA methods that overlook fairness concerns.
>
>
> > **Q5:** Were there any cases in the experiments where the framework significantly sacrificed accuracy to achieve fairness?
>
> **Response:**   Our empirical results show that our framework maintains accuracy while enhancing fairness. In NHANES, MF-CCA and SF-CCA correlations are only 0.5% and 1% lower than CCA in 2 dimensions. Max disparity drops by 26.4% and 29.5% compared to CCA, indicating fair enhancement with minimal accuracy loss across dimensions (Table 1).
>
> Figure 3 in the attached PDF further validates this benefit. Each dataset panel displays two cases for specific dimensions. MF-CCA and SF-CCA columns show changes in correlation ($\rho$), max disparity ($\Delta_{\max}$), and disparity sum ($\Delta_\Sigma$). Pearson correlation $\rho$ change is slight, while $\Delta_{\max}$ and $\Delta_\Sigma$ changes are substantial, signifying fairness improvement without significant accuracy sacrifice.

---

### Official Review · Reviewer_MdZ1 · 2023-07-06

**Soundness:** 3 good
**Presentation:** 3 good
**Contribution:** 3 good
**Rating:** 7
**Confidence:** 4

**Summary:**

This paper investigates the concept of Fair CCA, focusing on addressing the potential bias that arises when analyzing the relationship between two sets of variables using CCA, a widely utilized statistical technique. The conventional application of CCA fails to account for the impact of sensitive attributes like gender or race, leading to potential biases. In response, this study aims to bridge this gap by integrating fairness principles into CCA. The authors introduce the fairness issue within the context of CCA and propose two distinct methods to tackle it: a multi-objective approach and a single-objective approach, each offering unique strengths. The effectiveness of the proposed methods is substantiated through empirical and theoretical analyses, confirming their value in addressing the fairness concerns in CCA.


**Strengths:**

1. The problem addressed in this paper holds significant importance. Given the increasing influence of machine learning algorithms and methods on individuals and society, it becomes crucial to delve into the study of fairness within this domain. By mitigating bias issues in machine learning, we can contribute to a more equitable outcome and benefit vulnerable groups. While numerous works have explored fairness in machine learning, the majority of them focus on the supervised learning scenario. In contrast, this paper ventures into uncharted territory by examining the fairness issue in CCA, an unsupervised learning approach. This unique perspective underscores the urgency and significance of studying fairness within the context of CCA.
2. The concepts and methods presented in this paper exhibit a high degree of novelty. To the best of my knowledge, this is the first study to explore fairness within the context of CCA. Fairness, being a multifaceted concept, encompasses various definitions. In the realm of supervised learning, researchers have proposed different definitions such as demographic parity, equalized odds, and group sufficiency. Therefore, establishing a practical and reasonable definition becomes crucial. This paper introduces the notion of fairness criteria through correlation disparity error, which takes into account both global and group-wise correlations. The resulting fairness definition is intuitive and reasonable. The incorporation of fairness as additional objectives (in the multi-objective framework) or constraints (in the single-objective framework) is accomplished seamlessly, aligning with the natural progression of the problem. Furthermore, the authors introduce the Riemannian manifold in their solutions, which promotes convergence and facilitates computation, thereby introducing a novel aspect to the research.
3. The paper exhibits good writing quality, characterized by clarity and soundness. It effectively guides readers through its content, ensuring easy comprehension from the motivation and definition to the methods and solutions. Notably, Figure 1 provides a clear and intuitive visualization that enables immediate understanding of the proposed method's functionality. The effectiveness of the methods is supported by robust experimental results on synthetic and real data. Furthermore, Figure 3 serves as a compelling validation of fairness, as it visually demonstrates the improved proximity between the two groups after the projection using the proposed methods. Overall, the paper is meticulously crafted, maintaining a high level of clarity and rigor throughout.



**Weaknesses:**

1. Limited discussion on multiple modalities: While CCA is not restricted to two modalities, the paper primarily focuses on this scenario. It would be beneficial to discuss a more general setting involving multiple modalities and computing correlations under the fairness setting.
2. Figure 4 lacks an obvious trend: The authors could consider including Figure 9 from the supplementary file in the main body, as it provides a more intuitive demonstration of the method's effectiveness.

**Questions:**

1. Clarification on "critical Pareto" (line 164): The definition of "critical Pareto" is unclear. It would be helpful to provide a more intuitive explanation for better understanding.
2. Elaboration on optimization problem (8) and steepest descent direction (lines 174-175): The statement regarding the optimization problem and steepest descent direction requires further elaboration to enhance clarity.

**Limitations:**

The method is currently limited to two modalities. Even though this is the most common scenario in CCA, it would be interesting to see how the method can be extended to more than two modalities.

Overall, this is a well-written and informative paper that makes a significant contribution to the field of machine learning. The proposed methods are novel and effective, and the experimental results are convincing. However, the method is currently limited to two modalities, and it would be interesting to see how it can be extended to more than two modalities.

---

> ### Author Rebuttal · Authors · 2023-08-09
>
> > **W1:** Limited discussion on multiple modalities: While CCA is not restricted to two modalities, the paper primarily focuses on this scenario. It would be beneficial to discuss a more general setting involving multiple modalities and computing correlations under the fairness setting.
>
> **Response:**  Thank you for your valuable suggestion. We will discuss extending F-CCA to multiple modalities or generalized scenarios in our paper as future work.  Existing multiset CCA methods mostly revolve around the concept of maximizing pairwise correlation sums across subsets. We intend to leverage this concept to accommodate multiple modalities within our framework. Yet, these extensions come with notable challenges. Effectively handling fairness, bias, and solving single/multi-objective optimization problems across multiple modalities or sets requires fresh formulations and algorithmic approaches. This entails optimizing correlation or generalized correlation among sets while factoring in fairness, stability, and computational complexity, particularly when dealing with high-dimensional data issues [[ZYC20](https://pubmed.ncbi.nlm.nih.gov/32592530/),[TT11](https://link.springer.com/content/pdf/10.1007/s11336-011-9206-8.pdf),[X68](https://psycnet.apa.org/record/473742008-115?doi=1)]. Hence, developing a fair Generalized CCA approach requires addressing numerical stability and algorithm design, which we consider as future research problems.
>
> > **W2:**  Figure 4 lacks an obvious trend: The authors could consider including Figure 9 from the supplementary file in the main body, as it provides a more intuitive demonstration of the method's effectiveness.
>
> **Response:** Thank you for your suggestion. We'll replace Figure 4 with Figure 9, illustrating aggregate disparities across multiple datasets with varying projected dimensions. In our discussion, we'll detail SF-CCA and MF-CCA in connection with Figure 9. Notably, CCA consistently exhibits greater disparities than our SF-CCA and MF-CCA, highlighting our framework's effectiveness.
>
>
> > **Q1:**  Clarification on "critical Pareto" (line 164): The definition of "critical Pareto" is unclear. It would be helpful to provide a more intuitive explanation for better understanding.
>
> **Response:** Thank you for your comment.   In multi-objective optimization, conflicting objectives are simultaneously optimized, leading to trade-offs between solutions. The Pareto front comprises solutions that can't be enhanced in one objective without worsening another, defining Pareto optimal solutions. Pareto critical solutions lie on this front, representing vital trade-offs between objectives and providing optimal choices with distinct trade-offs for decision-makers.
>
> In Section 3.3, we introduced the concept of critical Pareto points. A point is considered critical Pareto if the image of the gradient of $\mathbf{F}$ at that point does not intersect with positive $M$-dimensional real numbers. In other words, the solution $\mathbf{P}$, as obtained from the optimization problem $(8)$, satisfies $\max_{i \in M} \langle \nabla f_i(\mathbf{U}, \mathbf{V}), \mathbf{P} \rangle = 0$. In simpler terms, there are no feasible directions $\mathbf{P} \neq \mathbf{0}$ that can simultaneously satisfy $\langle \nabla \mathbf{F}_i(\mathbf{U}, \mathbf{V}), \mathbf{P} \rangle \geq 0$ for all objectives while having at least one objective $i$ with $\langle \nabla \mathbf{F}_i(\mathbf{U}, \mathbf{V}), \mathbf{P} \rangle > 0$.
>
> In summary, Pareto critical points ensure that no feasible directions can reduce the norm of the component-wise gradient. This concept is similar to the single-objective case, where critical points prevent feasible descent directions from reducing the gradient norm.
>
>
> > **Q2:**  Elaboration on optimization problem (8) and steepest descent direction (lines 174-175): The statement regarding the optimization problem and steepest descent direction requires further elaboration to enhance clarity.
>
>
> **Response:** Thank you for your question. We have addressed subproblem (8) efficiently in Appendix A2 and discussed the manifold retraction operator in Appendix A1. A reference to Appendix A will be added in the main text, along with a summary of the following discussions.
>
> We elaborate on subproblem (8) and its relation to the convergence of standard gradient descent in Euclidean space. The definition of Pareto critical on Manifold extends the classical critical condition $\nabla f (x) = 0$ for the single objective case on Euclidean space ($m = 1$) to vector optimization on Manifold. Indeed, following single-objective optimization, we are looking for the directions $\mathbf{P}_t$ such that $ \left\Vert\mathbf{P}_t\right\Vert \rightarrow 0$.  To achieve this goal, we follow existing literature on multi-objective optimization  [[FS20](https://link.springer.com/article/10.1007/s001860000043)], and first use Lemma 7 (appendix) to show that the unconstrained optimization subproblem (8) has a unique solution $\mathbf{P}_t$ on the manifold and can be expressed in closed form. Hence, the solution to subproblem (8) can be computed using a variety of minimax methods, and we found that the Goal Attainment Method [[G75](https://ieeexplore.ieee.org/document/1101105), [FP86](https://www.infona.pl/resource/bwmeta1.element.ieee-art-000004256366), [F86](https://www.sciencedirect.com/science/article/abs/pii/B9780080316659500122)]  gives a more stable and accurate solution for our subproblem (8).  Next, Lemmas 9-10 (appendix) demonstrate that $\mathbf{P}_t$ is a descent direction and offer an estimate for the function $\mathbf{F}$'s decrease along the solution of (8). These findings mirror the descent of the objective in the single objective case and are extensively explored in the optimization literature for gradient descent-type methods.

---

> > ### Comment · Reviewer_MdZ1 · 2023-08-20
> >
> > Thanks for the author's response. I would also maintain my initial score of accept.

---

> > > ### Author Response · Authors · 2023-08-20
> > > **Response to Reviewer MdZ1**
> > >
> > > Thank you for your time and effort in reviewing our paper.

---

### Official Review · Reviewer_kQSh · 2023-07-06

**Soundness:** 3 good
**Presentation:** 4 excellent
**Contribution:** 3 good
**Rating:** 7
**Confidence:** 4

**Summary:**

This paper addresses a fairness issue that arises in CCA, proposing a fair CCA that well trade-offs correlation disparity errors w.r.t. sensitive attributes against correlation w.r.t. global projection subspaces. It introduces two optimization frameworks (multi-objective and single-objective), then developing corresponding efficient algorithms based on the generalized Stiefel manifold together with convergence analysis. Experimental results both on synthetic and real datasets are provided to validate the theoretical findings.

**Strengths:**

S1. The paper is very well written. Many illustrations (e.g., Fig. 1) are greatly helpful in grasping the ideas.

S2. The proposed optimization frameworks are convincing, and the corresponding translation techniques enable the use of efficient algorithms in the manifold literature. In addition, the theoretical analysis (Theorems 3 and 4) provides convergence guarantees of the algorithms.

S3. Experimental results emphasize the efficacy of the proposed approach, and the discussions therein are insightful.


**Weaknesses:**

W1. The translations for efficient algorithms, (8) and (9), can further be detailed for those who are not familiar with the manifold literature.

W2. For Theorems 3 and 4, the proof sketches (or technical contributions if any) are preferred to be included in the main body.


**Questions:**

See Weakness in the above.

**Limitations:**

See Weakness in the above.

---

> ### Author Rebuttal · Authors · 2023-08-09
>
> > **W1:** The translations for efficient algorithms, (8) and (9), can further be detailed for those who are not familiar with the manifold literature.
>
> **Response:**  Thank you for your valuable comment. We have addressed the efficient solutions to subproblems (8) and (10) in Appendices A2 and A3, and discussed the manifold retraction operator used in these sections in Appendix A1. To provide detailed explanations on the subproblems, we will add a sentence in the main text referring readers to Appendix A.
>
> Regarding the translations for efficient algorithms for solving subproblems (8) and (10), we understand the importance of providing additional details, particularly for readers who may not be familiar with the manifold literature. Following your valuable suggestions, we will include comprehensive discussions about subproblems (8) to enhance clarity and provide necessary context. We note that since subproblem (10) can be regarded as a specific case of subproblem (8) with $M=1$, where the function $f$ is replaced by its regularized counterpart on a smooth manifold, we focus on the latter. We will primarily include a summary of the following discussions in the paper.
>
>
> We provide a more detailed explanation of the multi-objective subproblem (8) and its connection to the convergence of standard gradient descent on Euclidean space. Notably, the definition of Pareto critical on Manifold extends the classical critical condition $\nabla f (x) = 0$ for the single objective case on Euclidean space (i.e., $M = 1$) to vector optimization on Manifold.
>
> Indeed, following single-objective optimization, we are looking for the gradient-based directions $\mathbf{P}_t$ such that $ \left\Vert\mathbf{P}_t\right\Vert \rightarrow 0$.  To achieve this goal, we follow existing literature on multi-objective optimization  [[FS00](https://link.springer.com/article/10.1007/s001860000043)], and first use Lemma 7 (appendix) to show that the optimization subproblem (8) has a unique solution $\mathbf{P}_t$ on the manifold and can be expressed in closed form. Hence, the solution to subproblem (8) can be computed using a variety of minmax methods, and we found that the Goal Attainment Method [[G75](https://ieeexplore.ieee.org/document/1101105), [FP86](https://www.infona.pl/resource/bwmeta1.element.ieee-art-000004256366), [F86](https://www.sciencedirect.com/science/article/abs/pii/B9780080316659500122)]  gives a more stable and accurate solution for our subproblem (8). Then, we provide Lemmas 9-10 (appendix) to show that $\mathbf{P}_t$ is actually a descent direction and provides an estimate for the decrease of the function $\mathbf{F}$ along the solution of (8). These results correspond to the descent of the objective in the single objective case and are well-studied in the optimization literature for gradient descent-type methods.
>
>
> > **W2:** For Theorems 3 and 4, the proof sketches (or technical contributions if any) are preferred to be included in the main body.
>
> **Response:** Thank you for the invaluable suggestion. Fortunately, NeurIPS permits an extra content page for the camera-ready version, which allows us to incorporate the following proof sketch:
>
> We provide the proof of Theorem 3 in three steps:
>
> * **Step I**: We use Lemma 7 to show that the optimization subproblem (8) has a unique solution $\mathbf{P}_t $.
> * **Step II**:  We provide Lemmas 9-10 to show an estimate for the decrease of the function $\mathbf{F}$ along the solution of (8); that is  for any $\eta_t \geq 0$, we have
> $
> \mathbf{F}(\eta_t\mathbf{P}_{t}) \preceq \mathbf{F}(\mathbf{U}_t, \mathbf{V}_t) - ( \eta_t-L_F \eta_t^2/2) \left \Vert \mathbf{P}_t \right \Vert^2 \mathbf{1}_M.
> $
> This can be seen as an extension from single-objective manifold optimization [[Nicolas23](https://www.nicolasboumal.net/book/IntroOptimManifolds_Boumal_2023.pdf)]  to the multi-objective counterpart.
> * **Step III**:  Finally, summing both sides of the inequality in Step II for $t=0, 1, \ldots, T-1$, and using our step size condition, gives the desired result.
>
>
> The proof of Theorem 4 follows steps similar to Theorem 3:
>
> * **Step I**:  We use a single objective variant of Lemma 7 to establish the unique solution $\mathbf{G}_t$ for the optimization subproblem (10).
> * **Step II**:  A single objective variant of Lemmas 9-10 shows that for any $\eta_t \geq 0$, we have a descent inequality as follows:
> $
> f(\eta_t\mathbf{G}_{t}) \leq f(\mathbf{U}_t, \mathbf{V}_t) - ( \eta_t-L_f \eta_t^2/2)  \left \Vert \mathbf{G}_t \right \Vert^2.
> $
> * **Step III**:  Finally, by summing both sides of the inequality in Step II for $t=0, 1, \ldots, T-1$, and using our step size condition, we obtain the desired result.

---

### Author Rebuttal · Authors · 2023-08-09

We thank all the reviewers for their valuable comments and suggestions. We summarize all the most-concerned questions raised by the reviewers below and present **new experiments**, accompanied by detailed explanations for the corresponding **attached Figures 1-3**.

* Reviewer **kQSh**  has highlighted the need for more detailed explanations of the efficient algorithms utilized to solve subproblems (8) and (10).

* Reviewer **MdZ1**  has emphasized the importance of addressing multiple modalities for CCA, suggesting an extension of our fair CCA approach to handle such scenarios.

* Reviewer **1Xpv** has expressed concerns about the generalizability of our framework and its applicability to various machine learning techniques or domains.

* Reviewer **6yqF** has observed that our approaches are more time-consuming compared to standard CCA , particularly in optimizing multi-objective problems.

* Reviewers **1Xpv** and **6yqF** have raised concerns regarding the trade-off between correlation disparity error (fairness) and correlation (accuracy).


To address these concerns, we have taken the following actions:

**a1:** We have provided more comprehensive descriptions and explanations of efficient algorithms utilized to solve subproblems (8) and (10), streamlining them for greater clarity and self-consistency within the main body of the paper. Although these descriptions were previously covered in Appendices A.2 and A.3, we aim to enhance their integration into the main text.

      Further details are provided in the response to Reviewer kQSh.

**a2:** We have given discussions regarding the extension of our fair CCA framework to accommodate multiple modalities. This includes exploring the application of our approach to scenarios such as multiset CCA, while acknowledging the challenges posed by optimization, scaling, and numerical stability issues.

      Further details are provided in the response to Reviewer MdZ1.

**a3:**  We have elaborated on the generalizability of our optimization framework to diverse machine-learning techniques and domains. This includes its potential extension to more generalized CCA variations including kernel CCA, deep CCA, and multiset CCA as suggested by Reviewer MdZ1. We also explore its application in subsequent CCA-based tasks like clustering and classification. Our approach of enhancing fairness by encouraging global projection matrices to align centrally with local projection matrices on the smooth Manifold has broader applicability and can be extended effectively to enhance other methods such as fair PCA.

      Further details are provided in the response to Reviewer 1Xpv.

**a4:**   We have conducted additional experiments to assess the time complexity of our approach, focusing on its sensitivity to the number of samples ($n$) and features ($d$).

- The findings, illustrated in Figure 1 of the attached PDF, highlight that MF-CCA’s runtime is notably sensitive to these factors, contrasting with the relative stability of CCA and SF-CCA runtimes. This sensitivity stems from MF-CCA's multi-objective nature, which necessitates optimizing $2K+2$ matrix variables for $M$ component objectives. Yet, this complexity offers the advantage of minimizing hyperparameter ($\lambda$) tuning while still achieving strong outcomes.

- We would like to emphasize that a runtime comparison of our methods against baseline CCA is indeed included in Table 2 of the paper. Additionally, the runtime sensitivity concerning the number of subgroups ($K$) is discussed in Figure 11 of Appendix B.3.

In response to the valuable suggestion by Reviewer 6yqF, we have incorporated new figures in the attached file to enhance the comprehensiveness of our runtime-related analysis.

    Further details are provided in the response to Reviewer 6yqF.

**a5:**  We have conducted new experiments, expanded discussions, and provided insights into the trade-off between fairness and accuracy. We show that our methods maintain accuracy while enhancing fairness, as evidenced by percentage changes.

  - In Figure 12 of the appendix, we demonstrate how performance reacts to the trade-off parameter $\lambda$ using synthetic data. Increasing $\lambda$ improves fairness but may slightly reduce accuracy, a reasonable outcome given its role in fairness regularization. Notably, the accuracy decline is modest despite significant fairness gains, showcasing our methods' capability to balance accuracy and fairness for SF-CCA.

  - Extending our experiments to real datasets, we present analogous results in Figure 2 of the attached PDF. Additionally, Figure 3 of the attached PDF depicts the performance changes across various datasets, underscoring our methods' ability to enhance fairness without sacrificing accuracy to a significant degree.

Therefore, the combined analysis of attached Figures 2-3 along with Figure 12 in the appendix offers a comprehensive understanding of the trade-off between fairness and accuracy.

     Further details are provided in the responses to Reviewer 1Xpv and 6yqF.


**Remark:** A few cited references in the response are absent from our paper's reference list. We assure their inclusion in the final version.

We appreciate the chance to address any concerns and questions raised by the reviewers during the discussion period.

---

### Decision · Program_Chairs · 2023-09-21

**Decision:**

Accept (poster)

**Comment:**

The reviewers unanimously accepted the paper and agreed that this was a noteworthy contribution to algorithmic fairness. I encourage the authors to consider the reviewers' suggestions, particularly in terms fo clarifying the theoretical guarantees and motivating the choice of fairness metrics.